# Enhanced Ca²⁺-channeling complex formation at the ER-mitochondria interface underlies the pathogenesis of alcohol-associated liver disease

Themis Thoudam [1], Dipanjan Chanda [1,2], Jung Yi Lee[2], Min-Kyo Jung [3], Ibotombi Singh Sinam [4], Byung-Gyu Kim [5], Bo-Yoon Park[1], Woong Hee Kwon[2], Hyo-Jeong Kim[6], Myeongjin Kim[1,7], Chae Won Lim[4,7], Hoyul Lee[1], Yang Hoon Huh[6], Caroline A. Miller[8], Romil Saxena[9,10], Nicholas J. Skill[11], Nazmul Huda[12], Praveen Kusumanchi[12], Jing Ma[12], Zhihong Yang[12], Min-Ji Kim [13], Ji Young Mun[3], Robert A. Harris[14], Jae-Han Jeon[15], Suthat Liangpunsakul [12,14,16,18] ✉ & In-Kyu Lee [1,17,18] ✉

Ca²⁺ overload-induced mitochondrial dysfunction is considered as a major contributing factor in the pathogenesis of alcohol-associated liver disease (ALD). However, the initiating factors that drive mitochondrial Ca²⁺ accumulation in ALD remain elusive. Here, we demonstrate that an aberrant increase in hepatic GRP75-mediated mitochondria-associated ER membrane (MAM) Ca²⁺-channeling (MCC) complex formation promotes mitochondrial dysfunction in vitro and in male mouse model of ALD. Unbiased transcriptomic analysis reveals PDK4 as a prominently inducible MAM kinase in ALD. Analysis of human ALD cohorts further corroborate these findings. Additional mass spectrometry analysis unveils GRP75 as a downstream phosphorylation target of PDK4. Conversely, non-phosphorylatable GRP75 mutation or genetic ablation of PDK4 prevents alcohol-induced MCC complex formation and subsequent mitochondrial Ca²⁺ accumulation and dysfunction. Finally, ectopic induction of MAM formation reverses the protective effect of PDK4 deficiency in alcohol-induced liver injury. Together, our study defines a mediatory role of PDK4 in promoting mitochondrial dysfunction in ALD.

Mitochondrial dysfunction is one of the major factors involved in the pathogenesis of alcohol-associated liver disease (ALD)[1,2]. Altered morphology and defective mitochondria are observed throughout the spectrum of ALD, from steatosis, hepatitis, and cirrhosis[3–5]. Primarily, alcohol is metabolized in hepatocytes where mitochondria play a crucial role in converting the highly reactive alcohol intermediate, acetaldehyde to acetate via acetaldehyde dehydrogenase[6]. Mitochondria are an essential organelle involved in nutrient metabolism and adenosine triphosphate (ATP) synthesis to regulate cellular homeostasis[7]. However, deterioration of mitochondria function contributes to excessive intracellular reactive oxygen species (ROS) formation leading to the activation of stress signaling pathways in ALD[1,8,9]. Among the myriad pathways involved in ALD, mitochondrial Ca²⁺ accumulation has been linked to mitochondrial dysfunction[10–12]. However, the factors involved in promoting mitochondrial Ca²⁺ accumulation and dysfunction during the pathogenesis of ALD remain elusive.

Mitochondria-associated ER membrane (MAMs), an interface between mitochondria and endoplasmic reticulum (ER) with an approximate distance of 10-50 nm[13], serves as a major hotspot for various signaling pathways to regulate mitochondrial dynamics, autophagy, and inflammasome formation[14,15]. One of its crucial functions is to regulate $Ca^{2+}$ transport from the ER to mitochondria to support mitochondrial metabolism and respiration. In the MAM, $Ca^{2+}$ transfer is regulated by the formation of the MAM $Ca^{2+}$-channeling (MCC) complex, consisting of glucose-regulated protein 75 (GRP75), inositol-1-tri-phosphate receptor 1 (IP3R1), and voltage-dependent anion channel 1 (VDAC1)[16]. GRP75 is a functional tether which facilitates the ER-mitochondria interaction, and efficient $Ca^{2+}$ transport to the mitochondria by physically interacting with the ER membrane $Ca^{2+}$ efflux channel, IP3R1, and the outer mitochondrial membrane (OMM) protein, VDAC1[17]. An aberrant increase in MAM formation promotes mitochondrial $Ca^{2+}$ overload and dysfunction during hepatic insulin resistance[18,19]. However, the involvement of the MAM in alcohol-induced mitochondrial $Ca^{2+}$ accumulation, mitochondrial dysfunction, and alcohol-induced liver injury has not been explored.

Pyruvate dehydrogenase kinase 4 (PDK4), one of the four PDK isoenzymes, regulates pyruvate dehydrogenase (PDH) complex activity to control glucose and fatty acid oxidation in the mitochondria[20]. Enhanced hepatic PDK4 expression in non-alcoholic steatohepatitis patients was shown to be correlated with disease aggravation[21]. On the other hand, genetic ablation of *Pdk4* protects against obesity-induced hepatic insulin resistance and steatosis[21-23]. We reported previously that a sub-population of PDK4 localizes in the MAM and interacts with the MCC complex to regulate the MAM formation. In addition, we found that increase in PDK4 expression induced MAM formation and promoted fatty acid-induced mitochondrial dysfunction[24]. Conversely, PDK4 deficiency prevented fatty acid-induced mitochondrial dysfunction and insulin resistance via suppression of MAM[20,24]. However, the potential role of PDK4 and MAM formation on mitochondrial dysfunction in ALD has not been elucidated. Here, we aimed to determine the contribution of MAM in the pathogenesis of alcohol-induced mitochondrial dysfunction and investigate the upstream factors that may modulate MAM formation in ALD.

In this study, we found that ALD exhibits abnormal enhancement of MCC complex-dependent MAM formation alongside a notable increase in PDK4 expression. Henceforth, we identify that PDK4 phosphorylates a key MCC complex component, GRP75, at multiple sites to promote alcohol-induced mitochondrial $Ca^{2+}$ accumulation and dysfunction. Conversely, genetic ablation of PDK4 ameliorates mitochondrial dysfunction and protects against alcohol-induced liver injury. Overall, we demonstrate a potential mediatory role of PDK4 in the pathogenesis of ALD.

## Results

### ER-Mitochondria contacts are augmented in alcohol-associated liver disease

To study the effect of alcohol on MAM dynamics, we utilized a well-established murine model of ALD[25] (Fig. 1a). Alcohol largely affects the hepatocytes in the perivenous region[26], therefore, we first analyzed liver sections obtained from mice fed with a control diet (CD) or ethanol (EtOH)-containing diet (ED) by transmission electron microscopy (TEM) in the perivenous region (Supplementary fig. 1a). Mitochondrial perimeter was significantly reduced in ED-fed mice compared to CD controls, while no difference was observed in the ER perimeter (Fig. 1b–d) between these two groups. Interestingly, the percentage of mitochondria in close proximity to the ER (within 50 nm distance) was significantly higher in ED-fed mice (Fig. 1b, e). Morphologically, most of the mitochondria in ED-fed mice were ER wrapped, resulting in increased MAM length compared to controls (Fig. 1f, g).

Next, we asked whether alcohol also affects the MAM dynamics in the periportal region (Supplementary Fig. 1a). In contrast to the perivenous region, the mitochondrial perimeter was significantly increased in the periportal region, while the ER perimeter was marginally reduced in ED-fed mice when compared with CD-fed mice (Supplementary Fig. 1b–d). On the other hand, similar to the perivenous region, we observed a significant increase in the percentage of mitochondria adjacent to the ER in ED-fed mice compared to CD-fed mice (Supplementary Fig. 1b, e). However, no difference in MAM length was observed between CD and ED-fed mice (Supplementary Fig. 1f, g). These results revealed a differential effect of alcohol on mitochondria and ER ultrastructure in the perivenous and periportal hepatocytes. Notably, despite having a difference in the MAM length, the association between ER and mitochondria was enhanced in both regions, suggesting that alcohol promotes MAM formation but more prominently in the perivenous region.

To reconfirm our in vivo observations, we treated isolated primary hepatocytes and murine hepatocyte cell line, AML12 cells with various doses of EtOH and confirmed that 100 mM EtOH is the optimal dose for inducing Cyp2E1, a sensitive indicator for alcohol exposure (Supplementary Fig. 2a, b). Next, we assessed the MAM formation by confocal microscopy after immunostaining the ER and mitochondria with protein disulfide isomerase (PDI) and translocase of outer membrane 20 (TOM20), respectively. We observed that EtOH treatment led to extensive co-localization of ER and mitochondria (Fig. 1h, i). Taken together, our results suggest that EtOH promotes the MAM formation in both in vitro and in vivo ALD models.

### Hepatic PDK4 expression and the MAM $Ca^{2+}$-channeling complex formation is elevated in alcohol-associated liver disease

Next, to delineate the molecular mechanism underlying EtOH-induced MAM formation, we performed transcriptomic analysis of our previously reported RNA-Seq datasets (GEO155830) of CD and ED-fed mice liver[27] to examine the differential expression of genes that are known to play a role in the MAM formation by encoding MAM resident proteins[14-16,28]. Among the 97 MAM-associated genes (Supplementary Table 1), four genes were significantly upregulated in ED-fed livers (pyruvate dehydrogenase kinase 4 [*Pdk4*], transglutaminase 2 [*Tgm2*], cell death inducing P53 target 1 [*Cdip1*], and activating transcription factor 6 [*Atf6*]), whereas one was downregulated (oxysterol binding protein like 8 [Osbpl8]) (Fig. 2a). The expression of *Pdk4* was the highest among the 4 upregulated genes (Fig. 2a, b). To validate the RNA-Seq results, we performed qPCR and immunoblot analyses and confirmed the induction of hepatic PDK4 in ED-fed mice at the transcript and protein levels (Fig. 2c–e). To draw the clinical relevance, we analyzed PDK4 expression in liver sections of patients with ALD (Supplementary Table 2). We noted that both mRNA and protein expression of PDK4 was substantially higher in patients with ALD compared to healthy subjects (HS) (Fig. 2f–i). None of the other PDK isoforms showed any observable difference in expression in mice, except for hepatic PDK2 expression which was induced in ED-fed mice (Fig. 2c–e). Moreover, enhanced PDK4 expression correlated with an increase in PDH phosphorylation at the Ser300 (Fig. 2d, e). However, PDK2 gene and protein expression remained unchanged, whereas PDK1 and PDK3 protein levels were reduced in patients with ALD but not in the mRNA levels compared to HS (Fig. 2f–h). This indicated that among the PDK isoforms, hepatic PDK4 was the only isoform consistently upregulated in both ED-fed mice and patients with ALD.

We have previously reported that PDK4 regulates the MCC complex formation[24]. Therefore, we evaluated whether the upregulation of PDK4 expression promotes MCC complex formation in ALD. To that end, we performed subcellular fractionation on CD and ED-fed mice livers to determine the expression of MCC complex proteins, particularly in the MAM. We found an increase in the localization of PDK4 along with IP3R1, GRP75, and VDAC1 in the MAM fractions in ED-fed

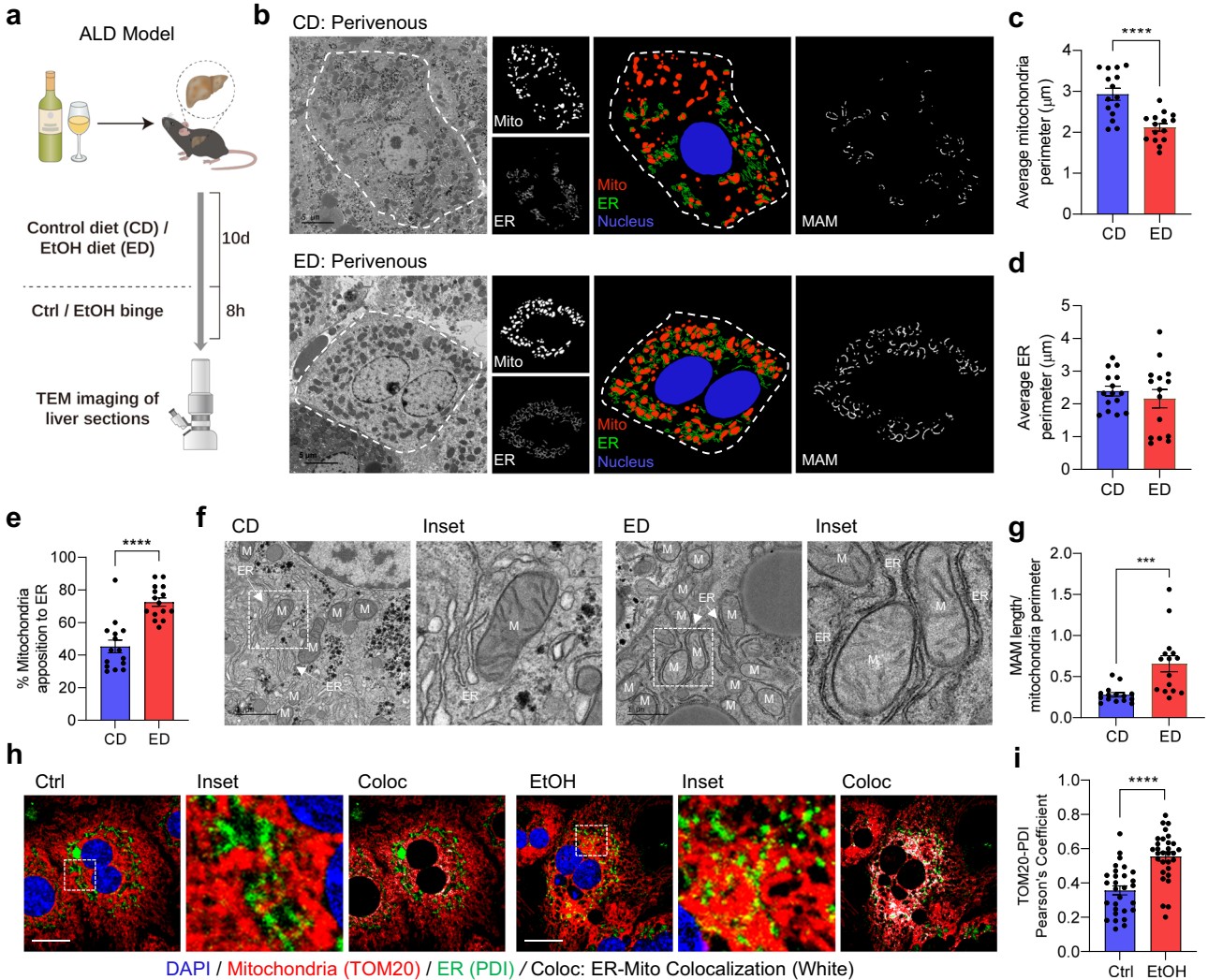

**Fig. 1 | Alcohol augments hepatic ER-mitochondria interaction. a** Graphical depiction of alcohol feeding model in mice. **b** Mitochondria and ER morphology in the perivenous region of control diet (CD) and EtOH diet (ED)-fed mice liver sections were visualized using TEM (Scale bars, 5 μm). The ER and mitochondria in the TEM images were reconstructed graphically to visualize MAM formation. (<50 nm distance between ER and mitochondria were considered as MAM). Mito: Mitochondria (red), ER: Endoplasmic Reticulum (green), Nucleus (blue). **c–e** Quantification of average mitochondria perimeter (**c**) and ER perimeter (**d**), and percentage of mitochondria apposition to ER (**e**), $n = 15$ microscopic fields each (number of mitochondria/ER analyzed, CD: 260/620 and ED: 390/1040 from 3 mice/group). **f** Magnified TEM images (Scale bars, 1 μm). M: Mitochondria, ER: Endoplasmic

Reticulum. **g** Quantification of MAM length to mitochondrial perimeter ratio, $n = 15$ microscopic fields each (number of mitochondria/ER analyzed, CD: 260/620 and ED: 390/1040 from 3 mice/group). **h** Confocal microscopy analysis of ER and mitochondria using PDI (Green) and TOM20 (Red) antibodies, respectively, in primary mouse hepatocytes cultured with or without 100 mM EtOH for 24 h (Scale bars, 20 μm). Colocalized portion of ER and Mitochondria is shown in white. **i** Quantification of co-localization between TOM20 and PDI (Ctrl, $n = 29$; EtOH, $n = 30$ microscopic fields with >200 cells from three independent experiments). All values are represented as mean ± SEM. ***$p < 0.001$; ****$p < 0.0001$ (Two-tailed unpaired $t$-test).

mice compared to CD-fed mice (Fig. 2j). Co-immunoprecipitation (co-IP) assays further revealed an increase in GRP75 interaction with PDK4, IP3R1, and VDAC1 but not with IP3R2 (the other IP3R isoform expressed in the liver) in ED-fed mice liver when compared to controls (Fig. 2k), confirming a direct physical association among the MCC complex proteins. Next, we evaluated the IP3R1-VDAC1 interaction by in situ proximity ligation assay (PLA), a method that detects the proximity between two proteins within a range of 40 nm, to evaluate MCC complex formation[29,30]. The specificity of IP3R1 and VDAC1 antibodies used for in situ PLA were validated by knocking down either IP3R1 or VDAC1 (Supplementary Fig. 2c–f). We found a significant increase in IP3R1-VDAC1 interaction upon EtOH treatment in primary mouse hepatocytes (Fig. 2l), AML12 cells (Supplementary Fig. 2g, h), and in the liver tissues of patients with ALD (Fig. 2m). Overall, these results demonstrate that MAM-associated PDK4 and MCC complex formation is elevated in ALD.

## PDK4 modulates MCC complex formation via GRP75 phosphorylation

As PDK4 is a serine/threonine kinase, we next examined whether PDK4 phosphorylates the MCC complex components. To that end, we performed immunoprecipitation using a phosphomotif antibody to detect the phosphorylation of MCC complex proteins. Interestingly, only GRP75 phosphorylation was induced upon EtOH treatment in primary mouse wildtype (WT) hepatocytes but not in the *Pdk4* knockout (*Pdk4*[−/−]) hepatocytes (Fig. 3a). As depicted in Fig. 2j, PDK4 and GRP75 are predominantly expressed in the mitochondria in addition to the MAM. To determine the site of interaction between PDK4 and GRP75, we performed in situ PLA in AML12 cells co-expressing FLAG-tagged PDK4, mitochondria target-blue fluorescence protein (mito-BFP), and ER-targeted SEC61B (*SEC61* Translocon Subunit Beta) tagged with GFP. In situ PLA blobs confirmed the interaction between GRP75 and PDK4 along with the individual MCC complex components

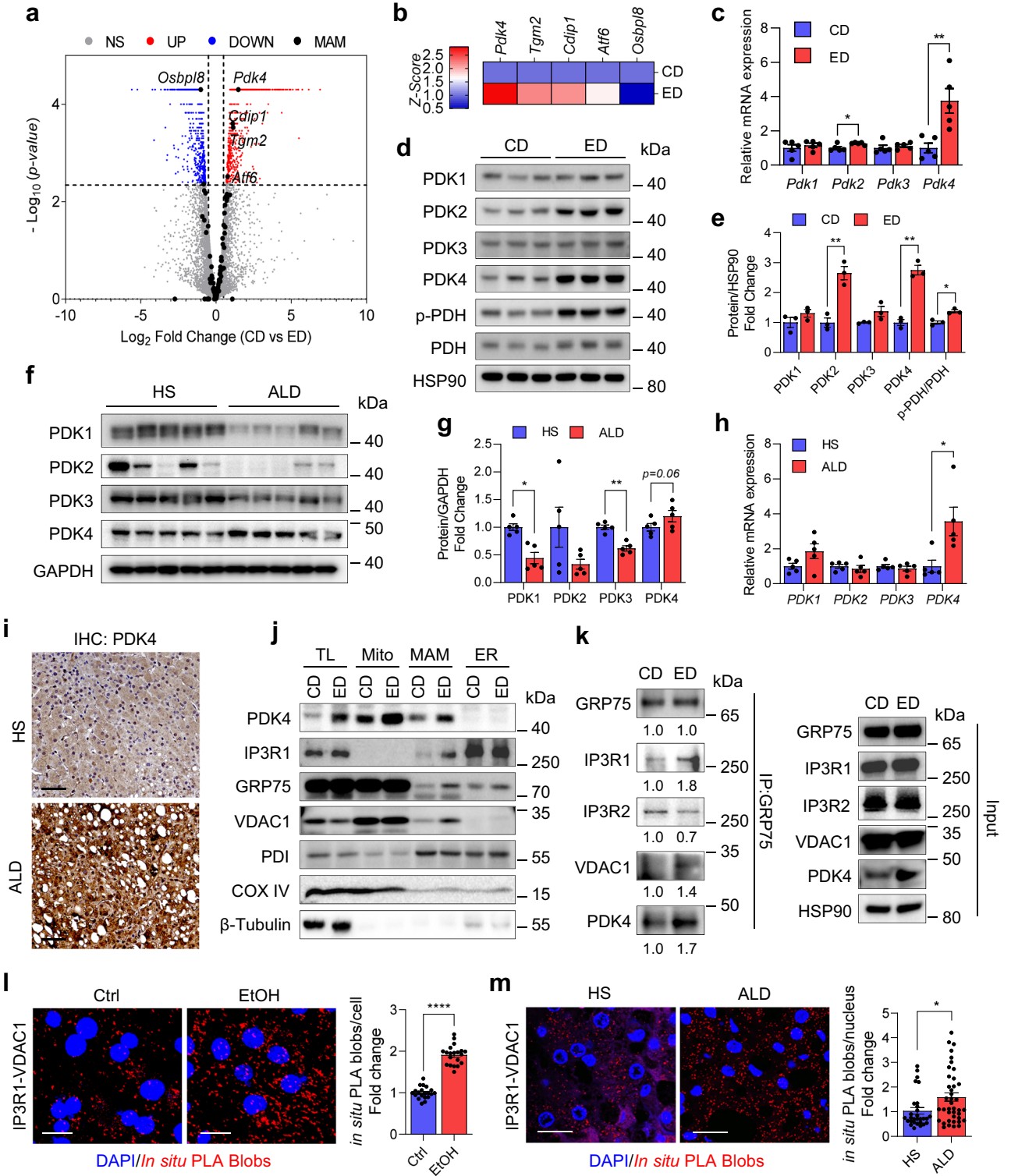

at the ER-mitochondria interface (indicated by white arrows in the inset) (Fig. 3b, c).

Next, we evaluated whether GRP75 is an immediate substrate of PDK4 by in vitro kinase assay using human GRP75 and PDK4 recombinant proteins. We found that PDK4 directly phosphorylates GRP75 (Fig. 3d). Further examination by mass spectrometry revealed 3 phosphorylation sites, corresponding to Thr120, Ser266, and Thr267 in the nucleotide binding domain (NBD) of human GRP75 (Fig. 3e and Supplementary Fig. 3a–c), which are highly conserved among the mammalian species (Fig. 3f). To gain further insight into the role of

GRP75 phosphorylation in MCC complex formation, we generated non-phosphorylatable mutants of GRP75 by replacing the phosphorylation sites with alanine (120 A, 266 A, and 267 A). The cDNA variants of GRP75 were then co-transfected with FLAG-tagged PDK4 in AML12 cells and analyzed by co-IP. PDK4-induced phosphorylation of GRP75 was markedly suppressed in 120 A, 267 A mutant proteins, and to a lesser extent by 266 A mutation. In addition, overexpression of PDK4 stimulated WT GRP75 interaction with IP3R1 but not with the mutant proteins (Fig. 3g, h). Of interest, we noticed that the PDK4-induced GRP75 interaction with VDAC1 was reduced by the 266 A and

**Fig. 2 | PDK4 is highly induced in human and mouse model of ALD. a** Gene expression of 97 known genes associated with MAM (Black dots) was analyzed from RNA-Seq (GEO155830) datasets of CD and ED-fed mice liver and represented as Volcano-plot. NS: Non-significant (Gray dots), UP: Significantly upregulated (Red dots) and DOWN: Significantly downregulated genes (Blue dots). **b** Heatmap showing the upregulated and downregulated MAM-associated genes in ED-fed mice compared to CD-fed mice. **c**–**e** mRNA ($n = 5$ mice/group) (**c**) and protein ($n = 3$ mice/group) (**d**-**e**) expression of PDK isoenzymes in CD and ED-fed mice liver. **f**–**h** protein (**f**, **g**) and mRNA (**h**) expression of PDKs in healthy subjects (HS) and alcohol-associated liver disease (ALD) human liver specimens ($n = 5$ each). **i** Representative image of immunohistochemical (IHC) analysis of PDK4 expression in the liver sections of HS and patients with ALD ($n = 5$ each) (Scale bars, 100 μm). **j** Immunoblot analysis of MCC complex proteins in the subcellular fractions isolated from CD and ED-fed mice liver. TL: tissue lysate, Mito: pure mitochondria,

MAM: mitochondria-associated ER membrane and ER: endoplasmic reticulum. **k** MCC complex formation was evaluated by co-immunoprecipitation (co-IP) using GRP75 antibody and immunoblotting of the indicated proteins (left), quantifications relative to CD are provided below each blot and inputs (right). **l** Representative confocal microscope images (left) and quantification (right) of in situ PLA showing the IP3R1-VDAC1 interactions in primary mouse hepatocytes treated with or without 100 mM EtOH for 24 h (Scale bars, 20 μm) ($n = 20$ microscopic fields each with >500 cells from three independent experiments). **m** Representative confocal microscope images (left) and quantification (right) of in situ PLA showing the IP3R1-VDAC1 interactions in HS and ALD human liver tissue sections (Scale bars, 20 μm) (HS, $n = 29$; ALD, $n = 40$ microscopic fields with >500 nuclei from 5 HS and ALD patient liver specimens). All values are represented as mean ± SEM, *$p < 0.05$; **$p < 0.01$; ****$p < 0.0001$ (Two-tailed unpaired t-test).

267 A mutations but not by the 120 A mutation (Fig. 3g, h), suggesting that the phosphorylation of GRP75 at T120 is dispensable for its interaction with VDAC1. Consistent with these results, in situ PLA showed that PDK4-stimulated IP3R1 interaction with WT GRP75 was markedly reduced with GRP75 mutants (Fig. 3i, j), confirming that PDK4 modulates MCC complex formation via GRP75 phosphorylation.

Next, we investigated the effect of PDK4-mediated MCC complex formation on mitochondrial Ca²⁺ uptake. An increase in MCC complex formation by over-expression of PDK4 (Fig. 3i, j) did not increase the basal mitochondria Ca²⁺ levels (Fig. 3k, l). However, stimulation of IP3R-mediated Ca²⁺ release by ATP significantly induced mitochondrial Ca²⁺ uptake in the cells co-expressing PDK4 and WT-GRP75, but not GRP75 mutants (Fig. 3k, m). Furthermore, GRP75 mutants did not affect the ATP-induced IP3R1-mediated ER Ca²⁺ release (Supplementary Fig. 4a), suggesting that reduction in mitochondrial Ca²⁺ uptake upon GRP75 mutation was not caused by decreased ER Ca²⁺ release. Taken together, our data indicate that PDK4-induced GRP75 phosphorylation promotes MCC complex formation and increases mitochondrial Ca²⁺ uptake.

### EtOH enhances MCC complex formation and mitochondrial Ca²⁺ accumulation via induction of GRP75 phosphorylation

To evaluate the effect of EtOH on PDK4-induced GRP75 phosphorylation and MCC complex formation, we performed co-IP assay in EtOH-treated AML12 cells overexpressing WT and mutant GRP75. EtOH markedly induced WT GRP75 phosphorylation, but not the mutants (Fig. 4a). Furthermore, EtOH treatment increased IP3R1 interaction with WT GRP75 but not with the GRP75 mutants (Fig. 4a). Consistent with the PDK4 overexpression (Fig. 3g), EtOH-induced VDAC1 interaction with WT GRP75 was compromised by 266 A and 267 A mutants but remained unaffected with the 120 A mutant (Fig. 4a). In situ PLA further revealed that EtOH-induced IP3R1-VDAC1 interaction was suppressed in GRP75 mutants overexpressing cells (Fig. 4b, c). Consequently, EtOH treatment increased the basal mitochondrial Ca²⁺ levels, which were further enhanced by ATP stimulation in WT GRP75 expressing cells but not in the mutants overexpressing cells (Fig. 4d–f). However, no difference in IP3R-mediated ER Ca²⁺ release was observed between the groups, except for a mild increase in 267 A mutant over-expressing cells (Supplementary Fig. 4b), suggesting that GRP75 mutation does not affect IP3R-mediated ER Ca²⁺ release. A sustained increase in mitochondrial Ca²⁺ level promotes excessive ROS generation causing mitochondrial dysfunction[31]. Furthermore, EtOH was shown to interfere with fatty acid oxidation[32] and the accumulation of incomplete fatty acid oxidation-derived metabolites can trigger mitochondrial Ca²⁺ accumulation[33]. We reasoned that EtOH-induced mitochondrial dysfunction may lead to excessive ROS generation and interfere with lipid metabolism leading to intracellular lipid accumulation. Indeed, EtOH treatment led to an increased mitochondrial ROS formation and enhanced lipid accumulation in WT GRP75 expressing cells, but not in the mutant expressing cells (Fig. 4g–i). Taken together, these data indicate that EtOH-induced MCC complex formation, and

subsequent mitochondrial Ca²⁺ accumulation lead to mitochondrial dysfunction and intracellular lipid accumulation.

### PDK4 deficiency prevents EtOH-induced mitochondrial Ca²⁺ accumulation and mitochondrial dysfunction

To examine the effect of PDK4 deficiency on EtOH-induced MAM formation and mitochondrial dysfunction in vitro, we first analyzed the MCC complex formation by in situ PLA. The proximity between the MCC complex proteins was significantly increased by EtOH in Pdk4⁺/⁺ hepatocytes but this was markedly suppressed in EtOH-treated Pdk4⁻/⁻ hepatocytes (Fig. 5a–d). Furthermore, analysis of the ER-mitochondria contacts by confocal microscopy revealed that EtOH-induced MAM formation was reduced in Pdk4⁻/⁻ hepatocytes (Fig. 5e, f). Similarly, knockdown of Pdk4 in AML12 cells suppressed EtOH-induced MAM formation (Supplementary Fig. 5a–c). As a consequence of reduced MAM formation in the absence of PDK4, EtOH-induced mitochondrial Ca²⁺ accumulation was significantly reduced in Pdk4⁻/⁻ hepatocytes (Fig. 5g), along with other measures of mitochondrial dysfunction, ROS accumulation (Fig. 5h), and mitochondrial membrane potential (MMP) (Fig. 5i). Notably, even in the absence of EtOH, the ROS level in Pdk4⁻/⁻ hepatocytes were considerably lower compared to Pdk4⁺/⁺ hepatocytes (Fig. 5h). Interestingly, evaluation of the oxygen consumption rate (OCR), a readout of mitochondrial function, revealed that the basal, ATP production-linked, and maximal OCR were significantly higher in Pdk4⁻/⁻ hepatocytes (Pdk4⁻/⁻-Ctrl) when compared to those in Pdk4⁺/⁺ hepatocytes (Pdk4⁺/⁺-Ctrl), (Fig. 5j, k). This suggests that PDK4 deficiency potentiates mitochondrial function as evidenced by the reduction in mitochondria ROS (Fig. 5h) and increased resting state ATP production-linked respiration. Additionally, EtOH significantly reduced OCR in Pdk4⁺/⁺ but not in the Pdk4⁻/⁻ hepatocytes (Fig. 5j, k). To confirm that these changes in mitochondrial function were not due to, alteration of mitochondrial OXPHOS or deregulation of antioxidant enzymes; Superoxide dismutase 2 (SOD2), catalase and glutathione peroxidases 4 (GPX4), we analyzed the expression of these proteins, but found no observable difference in OXPHOS complex or the antioxidant proteins expression in the absence or presence of EtOH between Pdk4⁺/⁺ and Pdk4⁻/⁻ hepatocytes. However, GPX4 protein level, an enzyme that protects against cellular lipid peroxidation[34] was reduced in EtOH-treated Pdk4⁺/⁺ hepatocytes but the expression was restored in Pdk4⁻/⁻ hepatocytes (Supplementary Fig. 5d). Together, these results indicate that PDK4 deficiency protects against EtOH-induced mitochondrial dysfunction via suppression of MCC complex formation.

### PDK4 deficiency suppresses MAM formation in ED-fed mice

Next, to verify our findings in vivo, we challenged wildtype (Pdk4⁺/⁺) and global Pdk4 knockout (Pdk4⁻/⁻) mice with either CD or ED, as depicted in Fig. 1a. No difference was observed in the body weight, daily food consumption and liver weight between Pdk4⁺/⁺ and Pdk4⁻/⁻ mice fed with CD or ED (Supplementary Fig. 6a–c). We found that ED-induced PDK4 upregulation was accompanied by increased GRP75

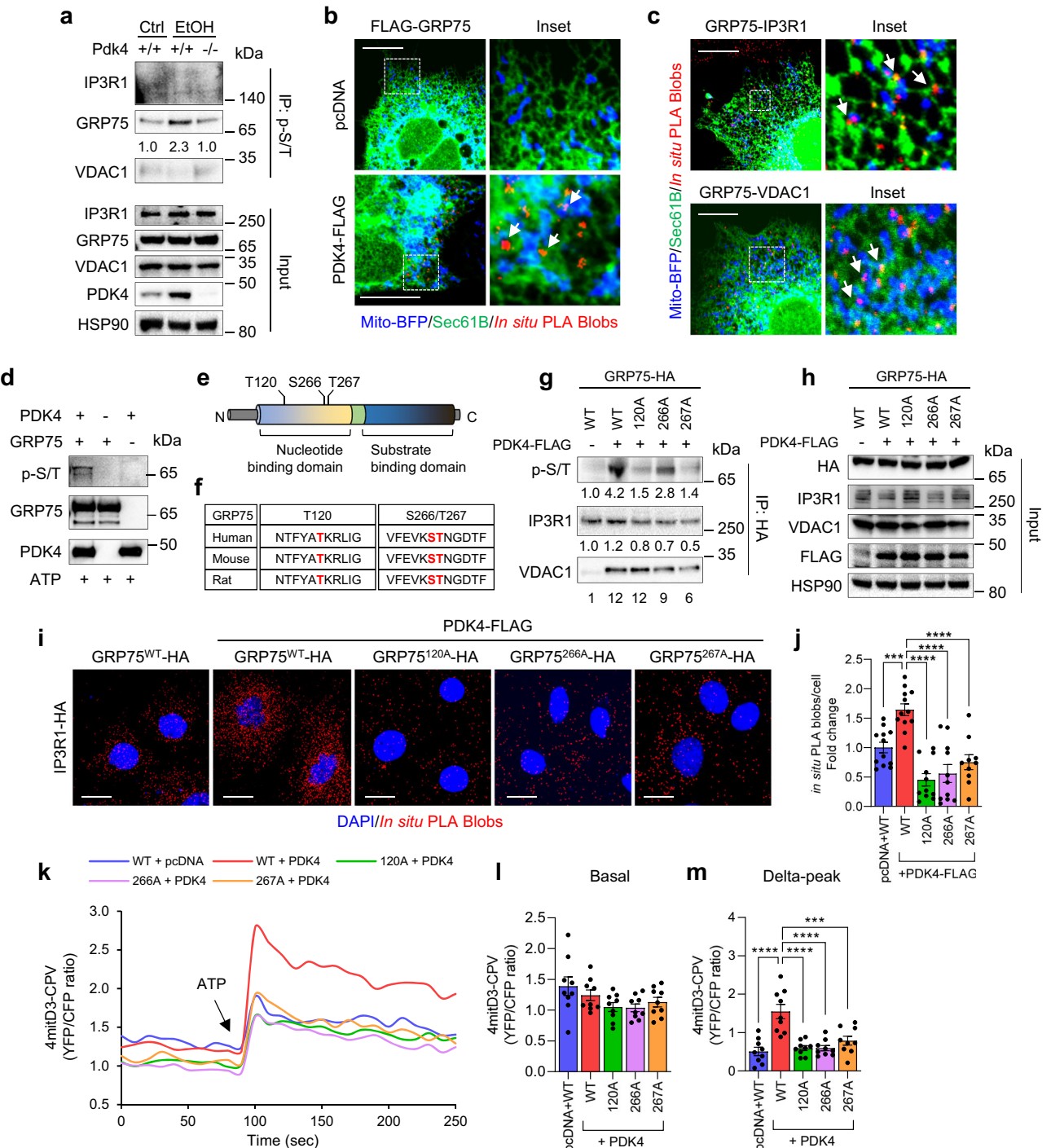

**Fig. 3 | PDK4 phosphorylates GRP75 to regulate MCC complex formation.**
**a** Phosphorylation of MCC complex proteins in *Pdk4⁺/⁺* or *Pdk4⁻/⁻* primary mouse hepatocytes treated with 100 mM EtOH for 24 h were analyzed by IP with phospho-serine/threonine (p-S/T) antibody. **b** PDK4-FLAG and GRP75 interaction (white arrows) was analyzed by in situ PLA in AML12 cells co-expressing pcDNA/PDK4-FLAG, mito-BFP (mitochondria) and Sec61B (ER) (Scale bars, 20 μm). **c** GRP75 and IP3R1/VDAC1 interaction (white arrows) was analyzed by in situ PLA in AML12 cells co-expressing mito-BFP and Sec61B (Scale bars, 20 μm). **d** In vitro kinase assay of recombinant GRP75 and PDK4 proteins in the presence of ATP and the phosphorylation of GRP75 was detected using p-S/T antibody by immunoblotting. **e** A model GRP75 protein structure highlighting the phosphorylation sites identified by mass-spectrometry. **f** Amino acid sequence comparison of GRP75 phosphorylation sites among the mammalian species. **g** GRP75 phosphorylation and its interaction with MCC complex proteins were analyzed by co-IP using HA antibody in AML12 cells co-overexpressing PDK4-FLAG and HA-tagged wild-type (WT) or phospho-

mutant GRP75 constructs. **h** Inputs of (**g**). **i** IP3R1 and exogenous GRP75 interaction was analyzed by in situ PLA in AML12 cells co-overexpressing PDK4-FLAG and HA-tagged WT or phospho-mutant GRP75 constructs. (Scale bars, 20 μm).
**j** Quantification of in situ PLA blobs of (**i**) (pcDNA+WT, $n = 12$; PDK4-FLAG + WT, $n = 12$; PDK4-FLAG + 120 A, $n = 11$; PDK4-FLAG + 266 A, $n = 11$; PDK4-FLAG + 267 A, $n = 10$ microscopic fields with >500 cells from 3 independent experiments).
**k** Measurement of mitochondrial Ca²⁺ level in AML12 cells (stably expressing 4mitD3-CPV) transfected with PDK4 and WT or phospho-mutant GRP75 constructs. The YFP and CFP fluorescence was recorded every 10 s. 100 μM ATP was injected at 90 s. **l, m** Quantification of basal (**l**) and ATP-stimulated delta-peak from the basal (**m**) values of (**k**). ($n = 9$ biological replicates from three independent experiments). All quantifications are represented as mean ± SEM, ***$p < 0.001$; ****$p < 0.0001$ (Ordinary one-way ANOVA, Dunnett's multiple comparisons test). Quantifications relative to control are provided below each blot.

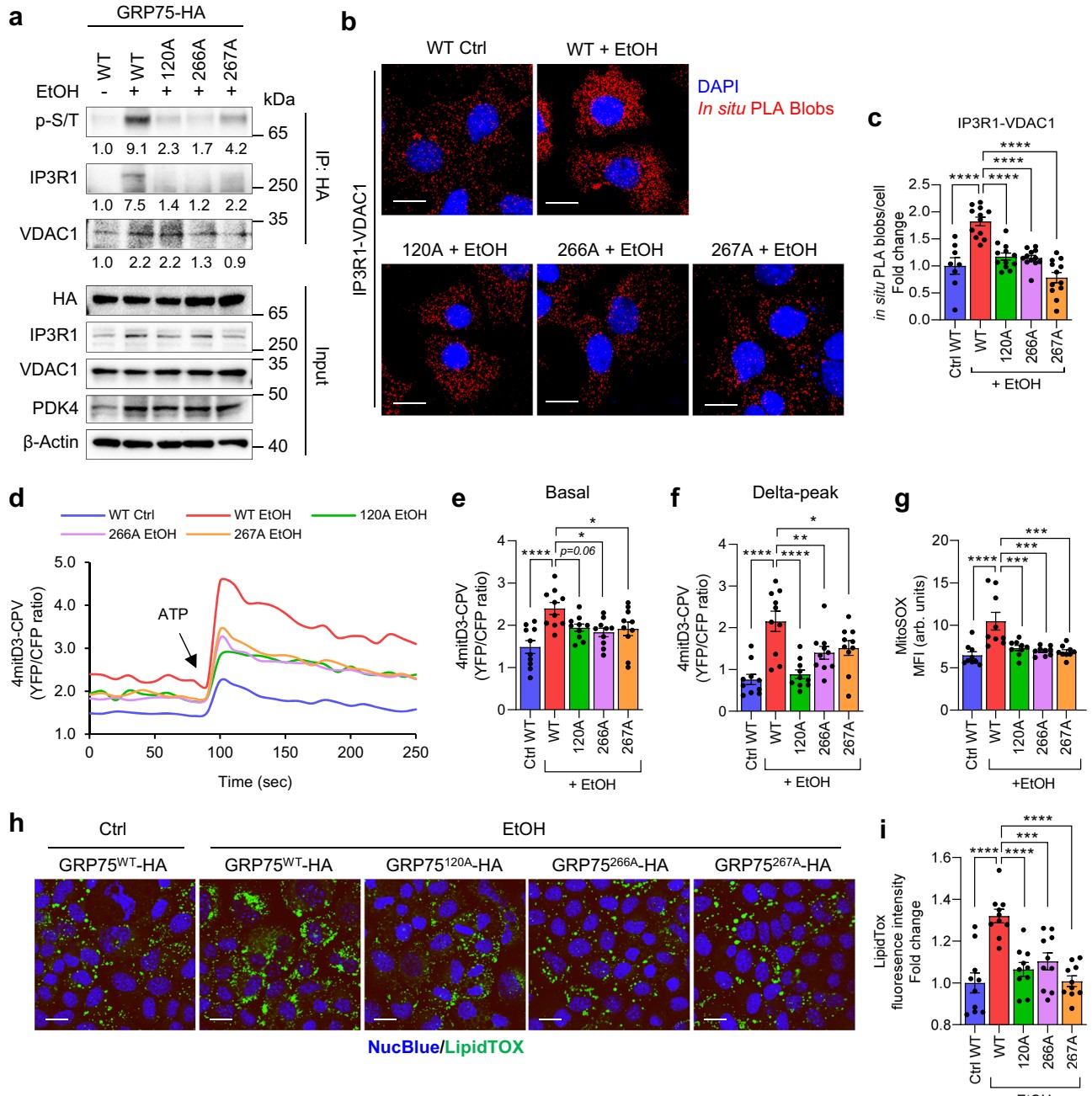

**Fig. 4 | GRP75 phospho-mutants suppress EtOH-induced MCC complex formation and mitochondrial Ca²⁺ accumulation. a** GRP75 phosphorylation and its interaction with IP3R1 and VDAC1 in AML12 cells overexpressing WT and phospho-mutant GRP75 treated with 100 mM EtOH for 24 h were analyzed by co-IP using HA antibody and immunoblotting. Quantifications relative to control are provided below each blot. **b** Analysis of IP3R1-VDAC1 interaction by in situ PLA in AML12 cells overexpressing WT and phospho-mutant GRP75 treated with EtOH (Scale bars, 20 μm). **c** Quantification of in situ PLA blobs of (**b**) (WT Ctrl, $n = 18$; WT/A120/A266/A267 + EtOH, $n = 12$ microscopic fields with >500 cells from 3 independent experiments). **d** Measurement of mitochondrial Ca²⁺ flux in AML12 cells (stably expressing 4mitD3-CPV) transfected with phospho-mutant GRP75 constructs and treated 100 mM EtOH for 24 h. The YFP and CFP fluorescence was recorded every 10 s and 90 s later 100 μM ATP was injected. **e, f** Quantification of basal (**e**) and ATP-stimulated delta-peak from the basal (**f**) values of (**d**) ($n = 10$ biological replicates from 3 independent experiments). **g** Analysis of Mitochondrial ROS level (mean fluorescence intensity; MFI) in AML12 cells overexpressing WT or phospho-mutant GRP75 and treated with 100 mM EtOH for 24 h using MitoSOX dye ($n = 9$ biological replicates from 3 independent experiments). **h** Intracellular neutral lipid accumulation was analyzed using the LipidTox dye in AML12 cells overexpressing WT or phospho-mutant GRP75 and treated with 100 mM EtOH for 24 h (Scale bars, 20 μm). **i** Quantification of fluorescence intensity of (**h**) ($n = 10$ microscopic fields with >500 cells from three independent experiments). All quantifications are represented as mean ± SEM, *$p < 0.05$; **$p < 0.01$; ***$p < 0.001$; ****$p < 0.0001$ (Ordinary one-way ANOVA, Dunnett's multiple comparisons test).

phosphorylation but this was markedly reduced in ED-fed *Pdk4⁻/⁻* mice (Supplementary Fig. 6d, e and Fig. 6a). Unexpectedly, PDK4 deficiency did not prevent ED-induced PDH phosphorylation (s300), which might be explained via increased PDK2 expression observed in both ED-fed *Pdk4⁺/⁺* and *Pdk4⁻/⁻* mice (Supplementary Fig. 6d, f, g). Next, we

evaluated the alterations in MCC complex proteins expression in MAM fractions isolated from CD or ED-fed *Pdk4⁺/⁺* and *Pdk4⁻/⁻* mice liver. PDK4 and other MCC complex proteins, as expected, were increased in the MAM fraction of ED-fed *Pdk4⁺/⁺* mice compared to the CD-fed controls, but not in the ED-fed *Pdk4⁻/⁻* mice (Fig. 6b–e). However, this

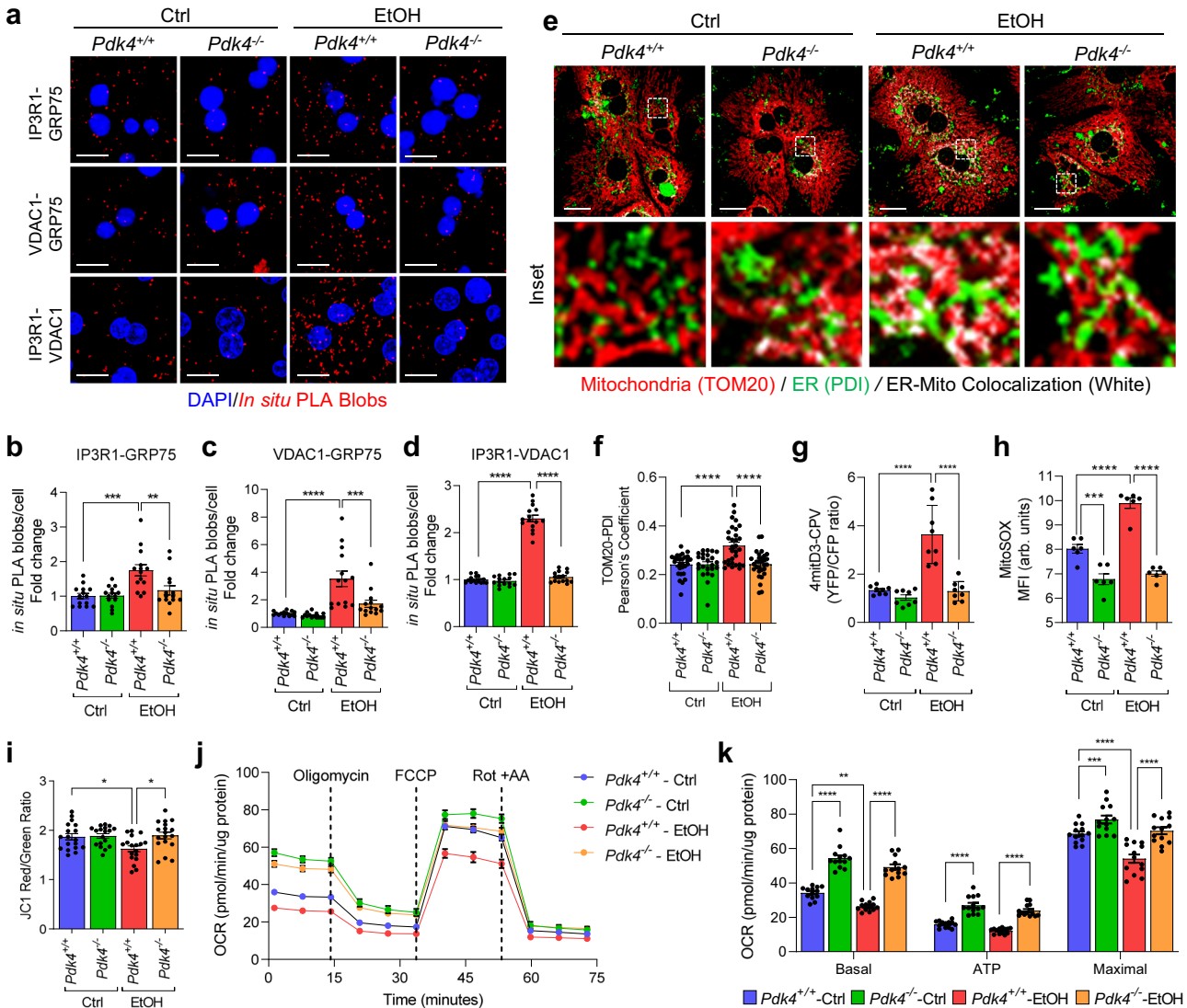

**Fig. 5 | PDK4 deficiency suppresses EtOH-induced MAM formation and mitochondrial dysfunction. a** Analysis of interaction between MCC complex proteins in *Pdk4*^+/+ and *Pdk4*^−/− primary mouse hepatocytes treated with or without 100 mM EtOH for 24 h by in situ PLA (Scale bars, 20 μm). **b–d** Quantification of (**a**) [IP3R1-GRP75 (**b**), $n = 14$; VDAC1-GRP75 (**c**), $n = 14$; IP3R1-VDAC1 (**d**), $n = 15$ microscopic fields with >300 cells from 3 independent experiments]. **e** Immunofluorescence analysis of ER and mitochondria using PDI (Green) and TOM20 (Red) antibodies in *Pdk4*^+/+ and *Pdk4*^−/− primary mouse hepatocytes treated with or without EtOH for 24 h. Colocalized portion of ER and Mitochondria is shown in white (Scale bars, 20 μm). **f** Quantification of TOM20-PDI colocalization of (**e**) (Ctrl *Pdk4*^+/+, $n = 28$; Ctrl *Pdk4*^−/−, $n = 24$; EtOH *Pdk4*^+/+ and EtOH *Pdk4*^−/−, $n = 30$ microscopic fields with >300 cells from 3 independent experiments). **g** Measurement of mitochondrial Ca^{2+} level

in *Pdk4*^+/+ and *Pdk4*^−/− primary mouse hepatocytes transduced with 4mitD3 adenovirus and treated with or without EtOH for 24 h ($n = 8$ biological replicates from three independent experiments). **h, i** Mitochondria ROS (mean fluorescence intensity; MFI) ($n = 6$ biological replicates) (**h**) and membrane potential ($n = 18$ biological replicates) (**i**) were measured using MitoSOX and the JC1 dye, respectively, in *Pdk4*^+/+ and *Pdk4*^−/− primary mouse hepatocytes treated with EtOH for 24 h (three independent experiments were performed). **j** Oxygen consumption rate (OCR) was measured by XF-analyzer. Rot: Rotenone, AA: Antimycin A. **k** Quantification of (**j**) ($n = 13$ biological replicates from three independent experiments). All quantifications are represented as mean ± SEM, *$p < 0.05$; **$p < 0.01$; ***$p < 0.001$; ****$p < 0.0001$ (Two-way ANOVA, Tukey's multiple comparisons test).

effect of PDK4 deficiency was not contributed by alterations in the total expression of MCC complex proteins (Supplementary Fig. 6h–k). Furthermore, TEM analysis of perivenous hepatocytes revealed the ED-induced increase in MAM length was significantly reduced in ED-fed *Pdk4*^−/− mice (Fig. 6f, g). Lastly, we evaluated the effect of ED on mitochondrial DNA content, the protein levels of mitochondrial OXPHOS complex and antioxidant enzymes. No significant difference in mitochondrial DNA content (Supplementary fig. 7a) and mitochondrial OXPHOS protein expression was observed in EtOH-treated hepatocytes or ED-fed mice liver compared to controls, except for Complex IV [cytochrome c oxidase subunit I (MT-CO1)] which was increased in ED-fed *Pdk4*^+/+ mice but not in ED-fed *Pdk4*^−/− mice (Supplementary Fig. 7b, c). Previously, MT-CO1 overexpression was

shown to promote ROS formation[35]. Interestingly, in line with the observation in vitro (Supplementary Fig. 5d), among the antioxidant enzymes, GPX4 protein expression was significantly reduced in ED-fed *Pdk4*^+/+ mice compared to CD-fed *Pdk4*^+/+ mice but not in ED-fed *Pdk4*^−/− mice (Supplementary fig. 7b, d–f), indicating that PDK4 deficiency may prevent alcohol-induced GPX4 depletion. Together, these results demonstrate that PDK4 deficiency suppresses alcohol-induced MAM formation in vivo.

## PDK4 deficiency ameliorates alcohol-induced liver injury by suppressing MAM formation

Next, we sought to examine whether PDK4 deficiency could ameliorate alcohol-induced liver injury. To this end, we examine the serum

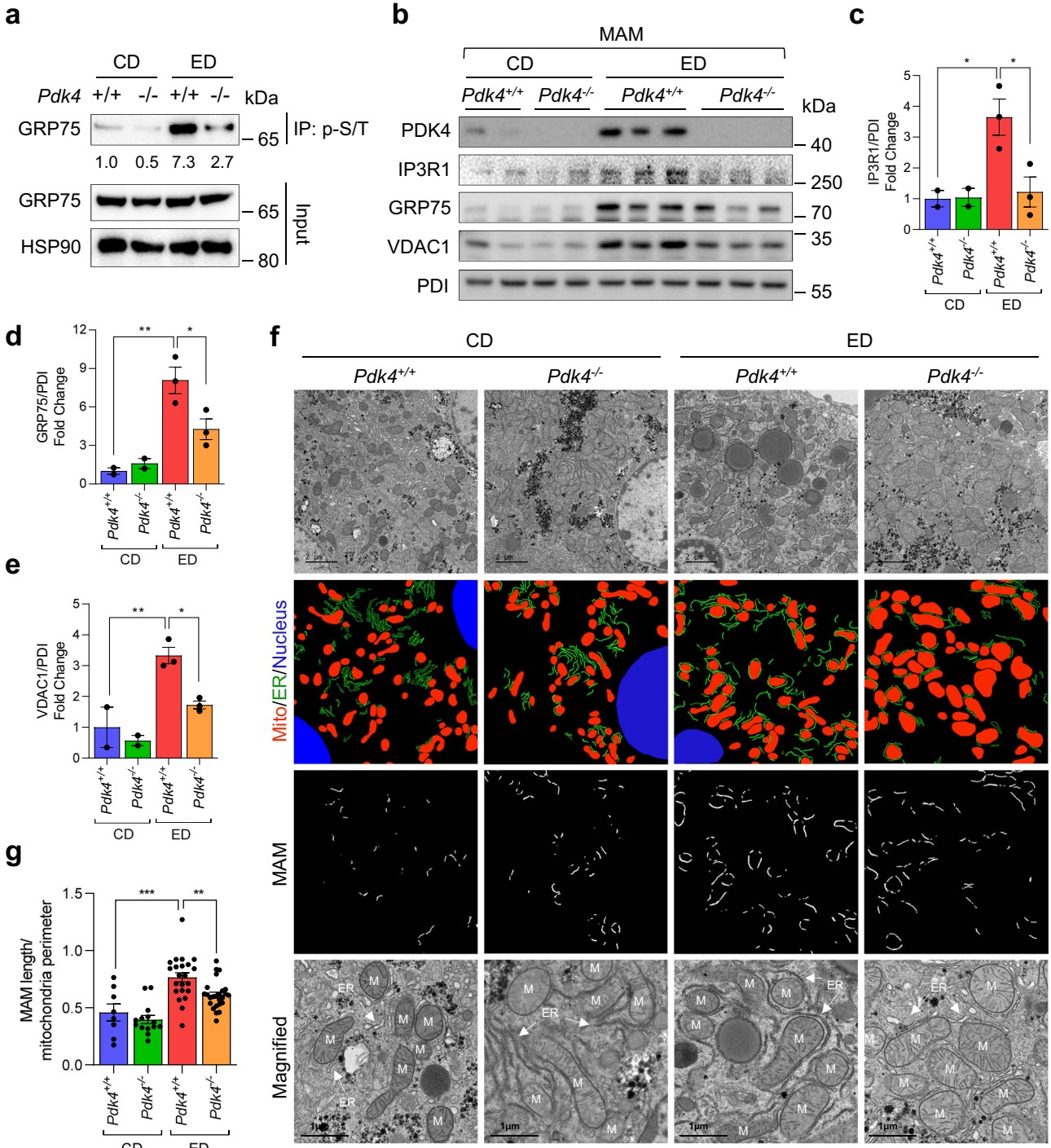

**Fig. 6 | Genetic ablation of PDK4 prevents alcohol-induced MAM formation in vivo. a** GRP75 phosphorylation in CD/ED-fed mice liver isolated from *Pdk4^+/+* and *Pdk4^-/-* mice was evaluated by IP with p-S/T antibody (representative blot of *n* = 3 mice/group). Quantifications relative to control are provided below each blot. **b–e** Immunoblot analysis of the MCC complex proteins in MAM fractions isolated from CD/ED-fed *Pdk4^+/+* and *Pdk4^-/-* mice livers (**b**), and quantifications (**c–e**) (CD *Pdk4^+/+*/*Pdk4^-/-*, *n* = 2; ED *Pdk4^+/+*/*Pdk4^-/-*, *n* = 3 mice/group). **f** MAM formation in the perivenous region of CD/ED-fed *Pdk4^+/+* and *Pdk4^-/-* mice liver sections were analyzed by TEM imaging [Scale bars, 2 μm (top panel) and 1 μm (magnified)]. The ER and mitochondria in the TEM images were reconstructed graphically to visualize MAM formation. Mito: Mitochondria (red), ER: Endoplasmic Reticulum (green), Nucleus (blue). **g** Quantification of MAM length to mitochondrial perimeter ratio, CD *Pdk4^+/+*, *n* = 8 (with 250 mitochondria); CD *Pdk4^-/-*, *n* = 14 (with 670 mitochondria); ED *Pdk4^+/+*, *n* = 22 (with 990 mitochondria); EtOH *Pdk4^-/-*, n = 25 (with 800 mitochondria) microscopic fields from 3 mice/group. All quantifications are represented as mean ± SEM, *$p < 0.05$; **$p < 0.01$; ***$p < 0.001$ (Two-way ANOVA, Tukey's multiple comparisons test).

aspartate aminotransferase (AST) and alanine aminotransferase (ALT) levels, indicators of hepatic injury. AST and ALT levels were markedly elevated in ED-fed *Pdk4^+/+* mice when compared to pair-fed *Pdk4^+/+* controls but significantly reduced in ED-fed *Pdk4^-/-* mice (Fig. 7a, b). Furthermore, ED-induced hepatic lipid accumulation (Fig. 7c) and

triglyceride content were significantly decreased in PDK4 deficient mice (Fig. 7d), indicating that genetic ablation of PDK4 protects against the development of alcohol-induced liver steatosis.

To gain a better understanding of the cell-type-specific function of PDK4, we thus generated hepatocyte-specific *Pdk4* knockout mice

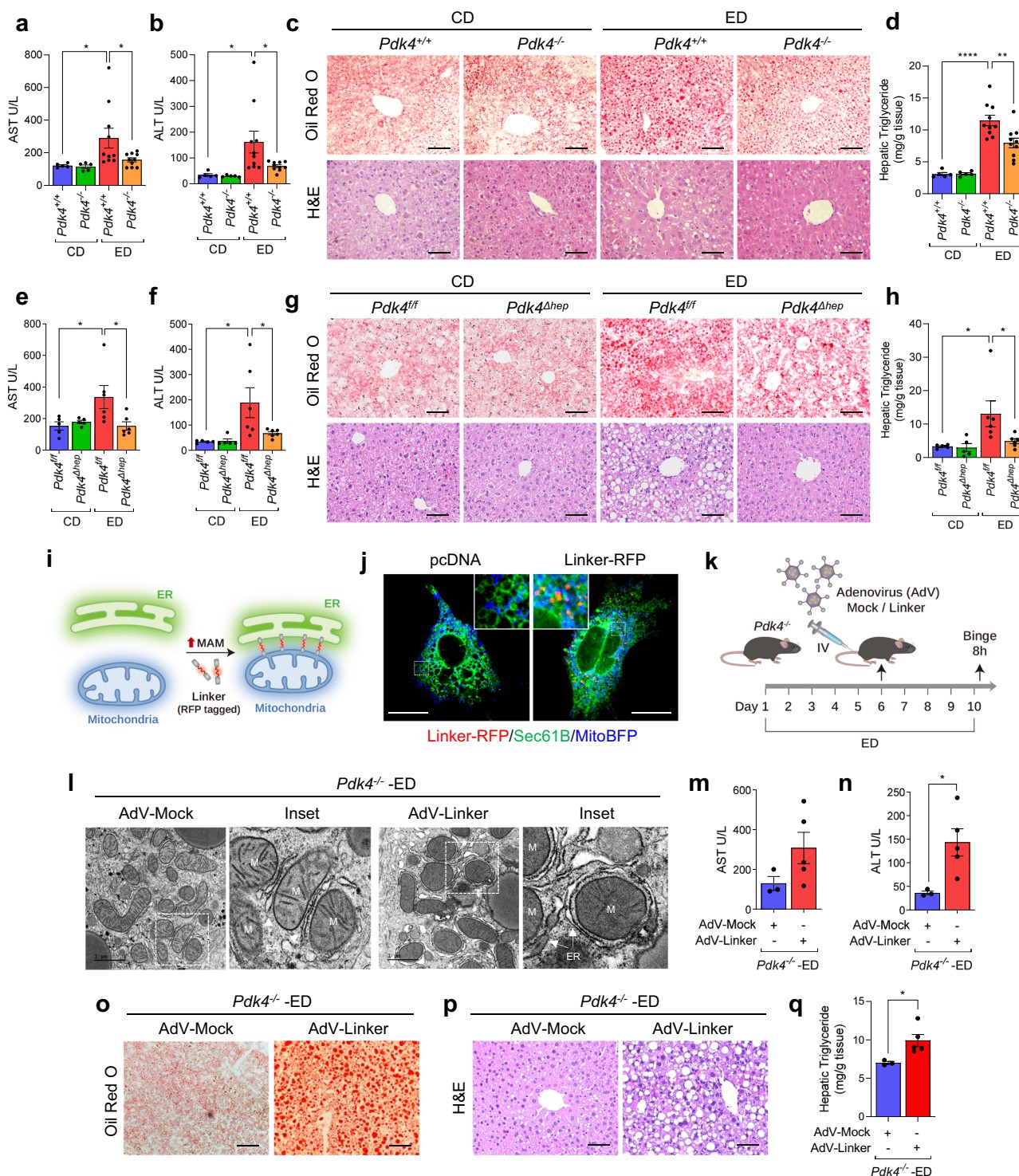

**Fig. 7 | Whole-body or hepatocyte-specific deletion of PDK4 alleviates alcohol-induced liver injury and is reversed by induction of MAM formation using Linker. a, b** Serum AST (**a**) and ALT (**b**) levels of CD/ED-fed *Pdk4+/+* and *Pdk4−/−* mice (CD *Pdk4+/+*/*Pdk4−/−*, *n* = 5; ED *Pdk4+/+*/*Pdk4−/−*, *n* = 10 mice/group). **c, d** Examination of lipid accumulation and histology by Oil-Red-O (top) and H&E (bottom) staining, respectively (**c**) (Scale bars, 50 μm), and triglyceride (TG) content (**d**) in CD/ED-fed *Pdk4+/+* and *Pdk4−/−* mice liver (CD *Pdk4+/+*/*Pdk4−/−*, *n* = 5; ED *Pdk4+/+*/*Pdk4−/−*, *n* = 10 mice/group). **e, f** Serum AST (**e**) and ALT (**f**) levels of CD/ED-fed *Pdk4f/f* and *Pdk4Δhep* mice (CD *Pdk4f/f*/*Pdk4Δhep*, *n* = 5; ED *Pdk4f/f*/*Pdk4Δhep*, *n* = 6 mice/group). **g, h** Oil-Red-O (top) and H&E (bottom) staining (**g**) (Scale bars, 50 μm), and TG content in CD/ED-fed *Pdk4f/f* and *Pdk4Δhep* mice liver (**h**) (CD *Pdk4f/f*/*Pdk4Δhep*, *n* = 5; ED *Pdk4f/f*/*Pdk4Δhep*, *n* = 6 mice/group). **i** Depiction of forced induction of MAM formation using Linker. **j** Targeting of Linker-RFP at MAM was analyzed by confocal microscopy in AML12 cells expressing mito-BFP and Sec61B with empty pcDNA/Linker-RFP (Scale bars, 20 μm). **k** Graphical demonstration of Mock/Linker adenovirus (AdV) delivery to hepatic tissue via intravenous (IV) injection in *Pdk4−/−* mice during ED-diet feeding period. **l** TEM images of AdV-Mock/AdV-Linker injected ED-fed *Pdk4−/−* mice liver sections (*n* = 3 mice/group) (Scale bars, 1 μm). **m, n** Serum AST (**m**) and ALT (**n**) levels of AdV-Mock/AdV-Linker injected ED-fed *Pdk4−/−* mice (AdV-Mock, *n* = 3; AdV-Linker, *n* = 5 mice/group). **o–q** Oil-Red-O staining (**o**) (Scale bars, 50 μm), H&E staining (**p**) (Scale bars, 50 μm), and TG content in (**q**) in AdV-Mock/AdV-Linker injected ED-fed *Pdk4−/−* mice liver (AdV-Mock, *n* = 3; AdV-Linker, *n* = 5 mice/group). All quantifications are represented as mean ± SEM, *\*p* < 0.05; *\*\*p* < 0.01; *\*\*\*\*p* < 0.0001 (Two-way ANOVA, Dunnett's multiple comparisons test and two-tailed unpaired *t*-test).

($Pdk4^{\Delta hep}$) by crossing $Pdk4$ floxed ($Pdk4^{f/f}$) with albumin-cre mice (Supplementary Fig. 8a) and fed them with CD and ED. No significant difference was observed in body weight, average food intake, and liver weight between the groups (Supplementary Fig. 8b–d). As expected, hepatic expression of PDK4 was significantly increased in ED-fed $Pdk4^{f/f}$ mice (Supplementary Fig. 8e, f). In congruence with global $Pdk4$ deletion, ED-fed $Pdk4^{\Delta hep}$ mice showed a reduction in serum AST and ALT levels, accompanied by lowered hepatic lipid accumulation and triglyceride content when compared to ED-fed control mice (Fig. 7e–h). Importantly, in line with our prior observations in $Pdk4^{-/-}$ mice (Supplementary Fig. 6d, f, g), ED-induced PDK2 expression and PDH phosphorylation levels remained unaltered in ED-fed $Pdk4^{\Delta hep}$ mice (Supplementary Fig. 8e, g, h), suggesting that the effect of $Pdk4$ deficiency in hepatic tissue produced by the whole body and hepatocyte-specific $Pdk4^{-/-}$ mice are indistinguishable. These findings confirm that PDK4 plays a significant role in the pathogenesis of ALD and that loss of PDK4 abrogates alcohol-induced hepatic steatosis and injury. Based on these findings, we tested the effect of PDK4 inhibition on EtOH-induced MAM formation and mitochondrial dysfunction using dichloroacetic acid (DCA). DCA treatment led to a significant reduction in EtOH-induced MCC complex formation, accompanied by decreased PDK4 expression and GRP75 phosphorylation (Supplementary Fig. 9a–c). Moreover, EtOH-induced elevation of mitochondrial $Ca^{2+}$ level, ROS generation and lipid accumulation were significantly decreased in DCA-treated cells (Supplementary Fig. 9d–g). These results suggest that inhibition of PDK4 might be a viable therapeutic route to treat ALD.

Finally, to understand the critical role of MAM formation in the pathogenesis of ALD, we overexpressed an ER-mitochondrial linker to enhance MAM formation and examine whether forced MAM formation could reverse the protective effect of PDK4 deficiency in ALD. Linker specificity was validated by confirming the localization of RFP at the MAM interface in AML12 cells labeled with mito-BFP and SEC61B-tagged GFP (Fig. 7i, j). Adenovirus constructs for mock control (AdV-Mock) and linker (AdV-Linker) were injected on the 6th day of ED feeding in $Pdk4^{-/-}$ mice by tail vein injection (Fig. 7k). Using this approach, we observed an increase in MAM formation in ED-fed $Pdk4^{-/-}$, confirmed by TEM imaging (Fig. 7l). No difference was observed in body weight and liver weight between the groups (Supplementary Fig. 10a, b). Linker introduction led to a significant increase in serum transaminase, notably ALT (Fig. 7m, n) and worsened hepatic steatosis in ED-fed $Pdk4^{-/-}$ mice (Fig. 7o–q), indicating a reversal of the phenotype upon induction of MAM formation. However, in line with a previous report[18], the administration of linker in absence of ED, despite having enhanced MAM formation failed to induce hepatic steatosis, gross morphological alteration, and TG accumulation (Supplementary Fig. 10c–f). This indicates that ectopic enhancement of MAM formation, at least in this experimental setting, is not sufficient to promote hepatic steatosis but requires a pathological trigger (in this case, ED) to promote hepatic steatosis. Taken together, our data indicate that PDK4-mediated MAM formation contributes to the pathogenesis of ALD (Fig. 8).

## Discussion

In the liver, almost all mitochondria are found to be in close contact with the ER within a distance of 100 nm also known as wrappER-associated mitochondria[36], whereas <50 nm distance between ER and mitochondria is the threshold for this contact to be considered as MAM[13]. MAM is a specialized contact site which plays an essential role in lipid synthesis, metabolite exchange, and $Ca^{2+}$ transport to regulate mitochondrial function[13,14], and it is dynamically regulated by the change in nutritional status of the cells[36]. An aberrant increase in $Ca^{2+}$ transport via MAM disturbs mitochondrial homeostasis and promotes mitochondrial dysfunction[18,28]. Dysregulation of mitochondrial $Ca^{2+}$ homeostasis causing mitochondrial dysfunction has been reported in

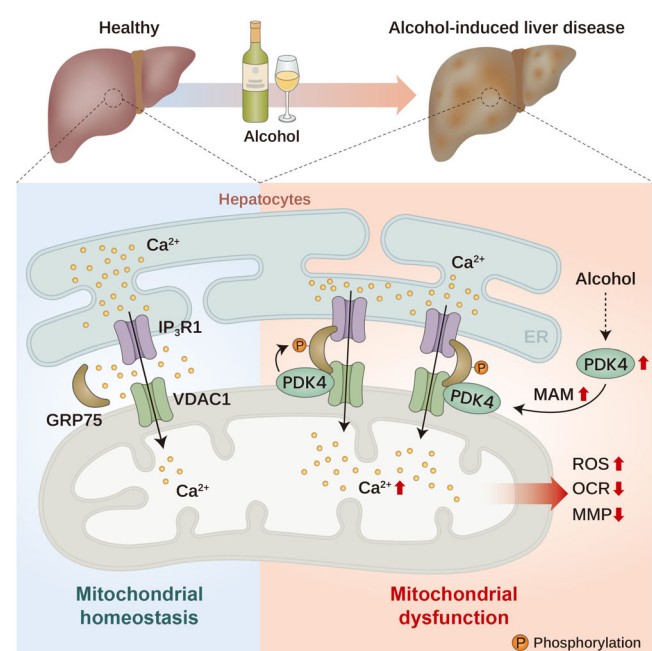

**Fig. 8 | Graphical illustration of PDK4-mediated $Ca^{2+}$-channeling complex formation at the ER-mitochondria contact site promotes mitochondrial dysfunction in alcohol-associated liver disease.** $Ca^{2+}$-channeling complex formation in MAM is augmented in alcohol-associated liver disease (ALD). An increase in PDK4 expression enhances the MAM $Ca^{2+}$-channeling complex formation via the phosphorylation of GRP75 to promote mitochondrial $Ca^{2+}$ accumulation and dysfunction in ALD. ROS: Reactive Oxygen Species, OCR: Oxygen Consumption Rate, MMP: Mitochondrial Membrane Potential.

ALD[10,11,37], however, the detailed mechanism underlying these findings remained unclear. GRP75 plays a critical role in the formation of MAM and also facilitates $Ca^{2+}$ transport by binding with IP3R1 and VDAC1[16,17]. The importance of GRP75 in regulating $Ca^{2+}$ transport between ER and mitochondria via the MCC complex is well established[16]. However, upstream regulation of the GRP75-mediated MCC complex formation has not been fully elucidated. We have previously demonstrated that PDK4 resides in the MAM and participates in the regulation of MCC complex formation[24]. This study provides evidence that PDK4 induces mitochondrial dysfunction in ALD by modulating MCC complex formation.

We demonstrated here that alcohol promotes MAM formation both in vivo and in vitro. In addition, we observed that an increase in the MAM formation is accompanied by enhanced MCC complex formation in both human and murine models of ALD. In depth transcriptomic analysis of MAM-associated genes revealed that PDK4 is highly elevated in the hepatic tissues of ED-fed mice and patients with ALD. Interestingly, transcriptional regulators of PDK4, namely, CCAAT enhancer binding protein beta (C/EBPβ)[38], and estrogen-related receptor γ (ERRγ)[39] were also shown to be upregulated in ALD[40,41], suggesting a potential involvement of these transcription factors in EtOH-mediated induction of PDK4 transcript level. Further investigation on the regulatory role of PDK4 on MCC complex formation identified GRP75 as the substrate for the modulation of the MCC complex. We found that PDK4 phosphorylates GRP75 at multiple sites, T120, S266, and T267. Using non-phosphorylatable mutants of these phosphorylation sites, we uncovered the mediatory role of PDK4 on alcohol-induced MCC complex formation and mitochondrial $Ca^{2+}$ accumulation. Of note, we observed that, unlike EtOH treatment, PDK4 overexpression alone failed to affect basal mitochondrial $Ca^{2+}$ levels in WT GRP75-expressing cells. However, upon stimulation of IP3R-mediated $Ca^{2+}$ release by ATP, both PDK4

　　　　　　　　　　　　　　　　　　　　　　　https://doi.org/10.1038/s41467-023-37214-4

overexpression and EtOH treatment strongly induced mitochondrial $Ca^{2+}$ uptake in WT GRP75-expressing cells, suggesting that PDK4 primes MCC complex formation but does not modulate ER $Ca^{2+}$ release at the basal state. $Ca^{2+}$ homeostasis plays a crucial role in maintaining cellular physiology; an excess level of $Ca^{2+}$ can activate ROS-generating enzymes leading to a deterioration of mitochondrial function[31]. We found that EtOH-induced mitochondrial $Ca^{2+}$ accumulation is accompanied by enhanced ROS formation. However, suppression of MCC complex formation by genetic ablation of PDK4 or the non-phosphorylatable mutants of GRP75 (120 A, 266 A, and 267 A) significantly reduced mitochondrial $Ca^{2+}$ levels and ROS formation. These data are in line with the previous findings that alcohol-induced mitochondrial $Ca^{2+}$ accumulation causes mitochondrial oxidative stress[10,12]. Our findings indicate that PDK4-mediated MCC complex formation initiates alcohol-induced mitochondrial $Ca^{2+}$ accumulation.

Mitochondria undergo rapid remodeling in response to stress as a part of the adaptive process; a prolonged mitochondrial stress with a compromised antioxidant system contributes to liver disease[7]. In congruence with the previous reports[37], we observed a decrease in mitochondrial respiration, MMP, and increased ROS formation which was accompanied by depletion of GPX4 protein level in EtOH-treated hepatocytes. Similarly, a significant reduction in GPX4 in ED-fed was observed compared to CD-fed control. However, this effect of EtOH was reversed upon genetic ablation of PDK4. These observations suggest that PDK4 may play a key role in driving the mitochondrial adaptive phase towards the maladaptive phase in ALD. Additionally, disruption in several cellular defense mechanisms, including autophagy and the redox system, have been implicated in the pathogenesis of ALD[42]. Mitochondrial quality control (MQC) is closely interlinked with these processes and plays a critical role in maintaining cellular homeostasis[7]. MAM plays a significant role in MQC by regulating mitochondrial fission[43]. Additionally, alcohol induces mitochondrial fission in hepatocytes;[37] in line with our observation, we found that ED reduced mitochondrial perimeter in the hepatocytes of the perivenous region, the primary site affected by alcohol[26]. We have recently demonstrated that PDK4 promotes stress-induced mitochondrial fission[44]. Together, these evidences suggest that PDK4 may play a role in mitochondrial fission in coordination with its role in MAM formation in ALD. Of note, in contrast to perivenous hepatocytes, the mitochondrial perimeter was increased in the periportal hepatocytes of ED-fed mice which correlates with an earlier report suggesting that alcohol promotes the formation of megamitochondria[45]. These findings suggest that the effect of alcohol on mitochondrial shape and sizes may vary in different zones of the liver but how hepatocytes in different zones react differently in response to alcohol needs further investigation. In this study, we have addressed how PDK4-mediated enhanced MCC-complex formation can play a part in promoting mitochondrial dysfunction in ALD. However, how PDK4 can affect autophagy or cellular redox system is currently unclear. Future studies are warranted to explore the role of PDK4 and MAM formation in these processes during the early adaptive phase of EtOH challenge as well as in chronic alcohol-induced liver injury.

Alcohol-induced mitochondrial dysfunction increases intra-hepatic lipid contents, via the inhibition of mitochondrial fatty acid oxidation[32], and the improvement in the mitochondrial function alleviates lipid accumulation[37,46]. PDK4 deficiency has been shown to suppress hepatic lipogenesis in non-alcoholic steatohepatitis[21,23]. Furthermore, overexpression of GRP75 promotes mitochondrial stress and lipid accumulation in hepatocytes[47]. We observed that alcohol-induced MCC complex formation and mitochondrial dysfunction were associated with intra-hepatic lipid accumulation in WT GRP75 but significantly less in GRP75 mutants expressing cells, suggesting a crucial role of PDK4 in the hepatocytes in the pathogenesis of ALD via regulating GRP75-mediated MCC complex formation. To determine

the functional implications of PDK4 in ALD pathogenesis, we employed two different PDK4 knockout models, a whole-body KO ($Pdk4^{-/-}$) and a hepatocyte-specific KO ($Pdk4^{\Delta hep}$). We found that both $Pdk4^{-/-}$ and $Pdk4^{\Delta hep}$ prevented alcohol-induced liver injury via the suppression of MAM. Additionally, a causal role of PDK4 on MAM-mediated hepatic steatosis in ALD is supported by the evidence that the forced induction of MAM formation abrogated the protective effect of PDK4 deficiency in alcohol-induced hepatic steatosis. Recent reports revealed that multiple proteins involved in lipid droplet bio-synthesis were found to be localized in MAM[48–50]. Therefore, the contribution of MAM formation in the context of lipogenesis in ALD warrants further investigation. Together, these findings indicate a critical role of the MAM in orchestrating mitochondrial dysfunction that induces liver injury and lipid accumulation during ALD pathogenesis. Of note, induction of PDK4 promotes fatty acid oxidation[51] and glycolysis[52] via inhibitory phosphorylation of PDH. Surprisingly, we found that PDK4 deficiency failed to suppress ED-induced PDH phosphorylation. This may be due to an increase in PDK2 expression observed in both ED-fed $Pdk4^{+/+}$ and ED-fed $Pdk4^{-/-}$ mice, as PDH is a common substrate of all PDK isoforms. Our results suggest that the protective effect of PDK4 deficiency on ED-induced mitochondria dysfunction, to a significant extent, is independent of the canonical function of PDK4. The direct involvement of MAM in regulating insulin signaling, mitochondrial dynamics, autophagy, inflammation, apoptosis and senescence pathways are well evident[15,53,54]. Interestingly, all these pathways were shown to play a role in the progression of ALD[37,55,56]. Collectively, these findings suggest that alcohol-induced MAM formation may not be limited to induction of mitochondrial dysfunction but it may also play a role in dictating other stress signaling pathways in ALD which requires further investigation.

There is, however, a major limitation in this study. We have predominantly verified our current hypothesis in male mouse model of ALD. Although this does not rule out the applicability of the mechanistic conclusions in female models, however, generalization outside the male models needs to be investigated in-depth to establish that the effect of PDK4 in alcohol-induced liver injury shows no prominent sexual dimorphism. In summary, we conclude that PDK4 is a key mediator of alcohol-induced liver injury, based on the following evidence. First, PDK4 deficiency prevents MAM formation in ED-fed mice. Second, the suppression of MAM formation in the absence of PDK4 prevents alcohol-induced mitochondrial $Ca^{2+}$ accumulation, mitochondrial dysfunction, and intracellular lipid accumulation. Third, the induction of ER-mitochondrial interaction in $Pdk4^{-/-}$ mice with a linker abrogated the protective effect on alcohol-induced liver injury. Together, our present study defines a causal role of PDK4 in the pathogenesis of ALD by mediating the MAM formation and that PDK4 is a potential therapeutic target for ALD.

## Methods
### Animal experiment
All experiments were conducted according to the guidelines established and approved by the Institutional Animal Care and Use Committee of Kyungpook National University and Indiana University. Wild-type ($Pdk4^{+/+}$), Pdk4 knockout ($Pdk4^{-/-}$) and hepatocyte-specific Pdk4 knockout ($Pdk4^{Hep\Delta}$) mice (generated by crossing $Pdk4$ flox mice with Albumin-Cre mice provided by Dr. Chul-Ho Lee, Korea research institute of bioscience and biotechnology, Daejeon, South Korea) of C57BL/6 J background were housed in a pathogens-free facility and maintained in standard housing conditions on a light/dark 12-h cycle at $22 \pm 2\,°C$.

Previous studies have reported that alcohol consumption causes hypothermia in human[57] and rodents maintained at 20-22 °C temperature[58,59]. In addition, alcohol feeding was shown to affect brown adipose tissue mass[60] and inter-organ cross talk influencing liver function[61,62]. Mice housed and maintained at thermoneutral conditions show greater non-alcoholic fatty liver disease and provide the

best way to investigate the cellular and molecular mechanisms underpinning alcohol liver disease pathogenesis[63]. Therefore, to rule out the influence of body temperature difference between control and alcohol-fed mice on liver function and its possible confounding effects on mitochondrial morphology and function, in this study, we induced ALD in mice under thermo-neutral conditions, i.e. 30 °C to investigate the potential role of MAM in alcohol-induced mitochondrial dysfunction.

All the experiments were performed in 10–12 weeks old male mice. ALD was induced in mice as described earlier[25] During the experiment, the mice were maintained in a 12 h light/dark cycle at 30 °C, and food consumption, body weight was monitored daily. Mice of the same age were acclimatized for 5 days with a control Lieber-DeCarli liquid diet (F1259SP, Bio-Serv) ad libitum. Then, the mice were fed with an ethanol diet (F1258SP, Bio-Serv) containing 5% (v/v) ethanol for 10 days. On the 11th day, each mouse received a single dose of Maltose Dextrin (3585; Bio-Serv) or 31.5% (v/v) ethanol (2701; Decon Labs) via oral gavage and was euthanized 8 hours (h) later. Collected serum and liver tissue samples were stored at −80 °C for analysis. For adenoviral delivery of ER-mitochondria linker in the liver, $1 \times 10^9$ plaque-forming units (PFU) mock/Linker adenovirus per mouse were injected intravenously. The ER–mitochondria linker tagged with red fluorescence protein (Linker-RFP) which would provide a 15-20 nm space between ER and mitochondrial outer membrane was provided by Dr. Hyun-Woo Rhee (Seoul National University, Seoul, South Korea).

## Human tissue sample
The use of human liver tissues of healthy controls and patients with ALD were approved by the Indiana University Institutional Review board under the study protocol 1712511018 and 1011004278. Written informed consent was obtained from all study participants.

**Healthy controls.** We obtained the liver tissues samples from the IRB-approved protocol at the Indiana University, Department of Surgery at Johns Hopkins Hospital (supported by the NIAAA, R24AA025017, Clinical resources for alcoholic hepatitis investigators), and Department of Pathology and Laboratory Medicine, Indiana University School of Medicine.

**Alcohol-associated liver disease samples.** Liver tissue samples from patients with alcohol-associated liver disease (alcoholic hepatitis) were provided by the Department of Surgery at Johns Hopkins Hospital. Baseline clinical and demographic data of alcohol-associated liver disease patients are provided in the Supplementary Table 2.

## Cell culture
Murine hepatocyte cell line, AML12 (CRL-2254; ATCC) was cultured in DMEM/F12 medium (11330-032; Gibco) supplemented with insulin-transferrin-selenium (41400-045; Gibco), 40 ng/mL dexamethasone (D1756; Sigma), 10% FBS (16000-044; Gibco) and 1X antibiotics (15140-122; Gibco) and all the experiments were performed between 4-20 passage number. Primary mouse hepatocytes were isolated from 8–10-week-old $Pdk4^{+/+}$ and $Pdk4^{-/-}$ mice by the collagenase perfusion method described previously[64] and cultured in M199 media (M4530; Sigma) supplemented with 23 mM HEPES (15630-080; Gibco), 10 nM dexamethasone (D1756; Sigma), 10% FBS (16000-044; Gibco) and 1X antibiotics. 24 h after plating, cells were treated with 100 mM Ethanol (459836; Sigma-Aldrich) in 0.5% FBS containing media for 24 h (ethanol-containing media was refreshed every 12 h). 2 mM Dichloroacetic acid (DCA) (347795; Sigma) was co-treated with EtOH for 24 h.

## Quantitative real-time PCR (qPCR)
Total RNA was isolated from tissues using QIAzol lysis reagent (79306; Qiagen) as described in the manufacturer's protocol. cDNA was synthesized from 2 μg of total RNA using a cDNA synthesis kit (K1622;

Thermofisher Scientific). qPCR was carried out using qPCR Master Mix (M3003L; New England Biolabs) in ViiA 7 Real-Time PCR system (Applied Biosystems). *36B4* or *GAPDH* was used for normalization. The primer sequences used in the study are listed in the Supplementary Table 3.

## Plasmid vector constructs and siRNA transfection
siRNAs targeting mouse *Pdk4* (Cat no. 1406883) [Sense 5'-CUCUAC UCUAUGUCAGGUU(dTdT)−3'; Antisense 5'-AACCUGACAUAGAGUAG AG(dTdT)−3'], *Itpr1* (Cat no. 1374716) [Sense 5'-CUGUAUGCGGAG GGAUCUA(dTdT)−3'; Antisense 5'-UAGAUCCCUCCGCAUACAG(dTdT) −3'], and *Vdac1* (Cat no. 22333-1) [Sense 5'-GACGGAUGAAUUCCAG CUU = tt(1-AS)−3'; Antisense 5'-AAGCUGGAAUUCAUCCGUC = tt(1-AA) −3'] and siControl (#SN-1003) were purchased from Bioneer (Daejeon, South Korea). Mitochondria-targeted blue fluorescence protein (mito-BFP) and ER-targeted Sec61B-GFP were provided by Dr. Hyun-Woo Rhee (Seoul National University, Seoul, South Korea). PDK4 with c-term FLAG tagged in pcDNA3 vector was provided by Dr. Robert A. Harris (Indiana University, IN, US). Human GRP75 with c-term hemag-glutinin (HA) tagged plasmid construct (HG16926-CY; Sino Biological) was used for generating non-phosphorylatable GRP75 mutants (T120-A, S266-A, and T267-A) using the QuickChange lightning site-directed mutagenesis kit (210518-5; Agilent technologies) and the mutations were confirmed by DNA sequencing. For transfection, cells were transfected with 100 nmol/L siRNA and plasmid DNA using RNAi Max reagent (13778150; Thermofisher Scientific) and Lipofectamine 2000 (11668019; Thermo Fisher Scientific), respectively, for 48 h and treated with 100 mM EtOH for 24 h.

## Co-immunoprecipitation (co-IP) and immunoblotting
Cells/tissues were lysed in RIPA lysis buffer with protease (04693132001; Roche) and phosphatase inhibitors (P0044; Sigma-Aldrich). Protein concentration was determined using BCA protein assay kit (23225; Thermofisher Scientific). For co-IP, cells/tissues were lysed in lysis buffer (50 mM Tris, 150 mM NaCl, 1% Triton X-100, 1 mM EDTA, and 10% glycerol) and 500 ug protein/sample was incubated with the indicated primary antibody for 3 h. A/G plus agarose beads (2003; Santa Cruz Biotechnology) were added and incubated over-night at 4 °C. The protein-bound beads were washed and boiled with SDS-PAGE sample buffer. Protein samples were then loaded, separated by SDS-PAGE, and transferred to a PVDF membrane (IPVH00010; Millipore). Membranes were incubated with indicated primary anti-bodies followed by incubation with horseradish peroxidase-linked secondary antibodies (anti-rabbit; Cell signaling, 7074 S or anti-mouse; R&D systems, HAF007; 1:2000 dilution) and detected using the iBright CL1500 Imaging System (Thermofisher Scientific). Primary antibodies used in the study are listed in the Supplementary Table 4.

## Subcellular fractionation
Subcellular fractions were performed in the freshly isolated liver tissue as previously described[65]. Briefly, minced tissue was homogenized with a Dounce homogenizer in Buffer 1 (225-mM mannitol, 75-mM sucrose, 0.5% BSA, 0.5-mM EGTA and 30-mM Tris–HCl pH 7.4) and the homo-genate was centrifuged at 740 g for 5 min. The supernatant was sepa-rated into crude mitochondrial pellet and ER containing supernatant by centrifugation at 9000 g for 10 min. The supernatant was collected for ER isolation, whereas, the crude mitochondrial pellet was resus-pended in buffer 2 (225-mM mannitol, 75-mM sucrose, 0.5% BSA and 30-mM Tris–HCl pH 7.4), and centrifuged at 10,000 g for 10 min. The pellet was resuspended in Buffer 3 (225-mM mannitol, 75-mM sucrose and 30-mM Tris–HCl pH 7.4), centrifuged at 10,000 g for 10 min, and the crude mitochondrial pellet was resuspended in the mitochondrial resuspension buffer (MRB) (250-mM mannitol, 5-mM HEPES; pH 7.4 and 0.5-mM EGTA). Mitochondria and MAM were separated with a percoll medium (225-mM mannitol, 25-mM HEPES; pH 7.4, 1-mM EGTA

and 30% Percoll; vol/vol) from the crude mitochondria by ultra-centrifugation at 95,000 g for 30 min. MAM layer and mitochondrial pellet was collected and resuspended in MRB buffer. MAM and mito-chondrial suspensions were centrifuged at 6300 g for 10 min. Pure mitochondrial pellet was collected, and MAM fraction was collected after pelleting by ultra-centrifugation at 100,000 g for 1 h. For ER isolation, the supernatant was centrifuge at 20,000 g for 30 min followed by ultracentrifugation at 100,000 g for 1 h. Isolated subcellular fraction was resuspended with lysis buffer containing protease and phosphatase inhibitors. Protein samples were resolved by Tris-glycine SDS PAGE and analyzed by immunoblot.

### Transmission electron microscopy (TEM)
Liver tissue sections were fixed with the fixative solution containing 2% paraformaldehyde and 2.5% glutaraldehyde, and then post fixed with 1% osmium tetroxide (OsO4) in ice for 2 h. The tissues were then washed with PBS and dehydrated in ethanol and propylene oxide series, embedded in Epon 812 mixture, and polymerized in an oven at 70 °C for 24 h. The sections acquired from polymerized blocks were collected on grids, counterstained with uranyl acetate and lead citrate, and examined with Bio-HVEM system (JEM-1000BEF at 1000 kV, JEOL, JAPAN) and Bio-TEM system (JEM-1400Plus at 120 kV, JEOL, JAPAN and Tecnai G2 at 120 kV, Thermo Fisher Scientific, Waltham, MA, USA). To map the perivenous and periportal region in the liver sections, a combination of low- and high-magnification EM imaging ranging from 38X to 5000X was applied to precisely locate the hepatocyte population in the perivenous and periportal region (within 50 μm from the central vein or portal vein border) as demonstrated in the Supplementary Fig. 1a. The architecture of mitochondria and ER in the EM images were reconstructed graphically using GIMP software (GNU image manipulation program, Gimp 2.10), and analyzed using ImageJ (NIH, Bethesda, MD, USA).

### Immunofluorescence
Cells were seeded on 12 mm glass coverslips in 24 well plates. At the end of the experiment, cells were washed with PBS, fixed with 4% paraformaldehyde, permeabilized (0.2% Triton X, 0.1 M glycine in PBS), and incubated with primary antibodies overnight at 4 °C. Next, the cells were washed, incubated with the secondary antibodies conjugated with fluorophores Alexa Fluor 488 (Invitrogen, A11001; 1:100 dilution)/Alexa Fluor 568 (Invitrogen, A11011; 1:100 dilution), and mounted using VECTASHIELD mounting medium containing DAPI (H1200). Images were captured using Olympus FluoView FV1000 confocal microscope (Olympus Imaging, PA, USA). Pearson co-efficient colocalization of ER and mitochondria was analyzed by ImageJ software with Fiji plugin (NIH, Bethesda, MD, USA). Primary antibodies used in immunofluorescence experiments were anti-rabbit TOM20 (Santa Cruz, sc-11415; 1:100 dilution) and anti-mouse monoclonal PDI (Abcam, ab2792; Clone RL90; 1:100 dilution).

### In situ Proximity ligation assay (PLA)
Protein-protein interactions were visualized using Duolink In Situ detection kit (DUO92002/4/7; Sigma-Aldrich) by following the manufacturer's protocol. Briefly, cells were cultured on 12 mm glass coverslips in 24 well plates. At the end of the experiment, cells were washed with PBS, fixed with 4% paraformaldehyde, and permeabilized (0.2% Triton X, 0.1 M glycine in PBS). Next, the cells were incubated with blocking solution, followed by overnight incubation with primary antibodies. The cells were then washed, probed with the secondary antibodies conjugated with oligonucleotide, ligated, and amplified. Preparations are then mounted using Duolink II mounting medium containing DAPI (DUO82040; Sigma-Aldrich), and the fluorescence signals were analyzed by ImageXpress Micro confocal and MetaXpress software (Molecular devices). Primary antibodies used in in situ PLA experiments were anti-rabbit VDAC1 (Abcam, ab15895; 1:100 dilution),

anti-mouse monoclonal IP3R1 (Santa Cruz; sc-271197; Clone E8; 1:50 dilution), anti-mouse monoclonal GRP75 (Santa Cruz, sc133137; Clone D9; 1:50 dilution), anti-rabbit IP3R1 (Invitrogen, PA1-901; 1:100 dilution), anti-rabbit FLAG (Cell signaling, #2368; 1:100 dilution) and anti-mouse HA (Santa Cruz, sc7392; Clone F-7; 1:50 dilution).

### In vitro kinase assay
In vitro kinase assay to detect protein phosphorylation was analyzed as previously described[66]. Briefly, PDK4 (provided by Dr. Nam-Ho Jeoung, Catholic University, Daegu, South Korea) and GRP75 (NBC1-18380; Novus biologicals) human recombinant proteins were mixed in 1:50 ratio in 1X kinase buffer (9802 S; Cell signaling) and incubated for 30 min at 30 °C in presence of 500uM ATP (A2383; Sigma). The reaction was terminated by the addition of 1X sample buffer and boiled. The samples are then analyzed by SDS-PAGE and immunoblotting.

### LC-MS/MS analysis
LC-MS/MS analysis was performed after de-stainning and excision of the target protein band from the Coomassie stained SDS-PAGE gel as follows. In short, the gel was sectioned into 10 mm sections, and in-gel digested with trypsin. The tryptic digest was separated through online reversed-phase chromatography using a reversed-phase peptide trap occlusion PepMap™100 (internal diameter, 2 cm length) and a reversed-phase analysis column PepMap™RSLCC,1875 (75 μm inner diameter, 15 cm length, 3 μm particle size), both from Thermo Scientific, followed by electrospray ionization at a flow rate of 300 nl min-1. Samples were eluted with a split gradient of 3-50% solution B (80% ACN with 0.1% FA) for 60 min and 50-80% solution B for 10 min, and the columns were washed in 100% solution B for 10 min. The chromatography system coupled in line with an Orbitrap Fusion Lumos mass spectrometer was operated in a data dependent mode with the 120,000 resolution MS1 scan (375-1500 m/z), an AGC target of 5e5, and a maximum injection time of 50 ms. Peptides above the threshold of 5e3 and a charge of 2-7 were selected for fragmentation through dynamic exclusion after 15 second and 10 ppm tolerance.

DATABASE SEARCHING- Charge state deconvolution and deisotoping were not performed. All MS/MS samples were analyzed using Sequest (Thermo Fisher Scientific, San Jose, CA, USA; version IseNode in Proteome Discoverer 2.4.1.15) and X! Tandem (The GPM, thegpm.org; version X! Tandem Alanine (2017.2.1.4)). Sequest was set up to search Uniprot-human.fasta (unknown version, 42230 entries) assuming the digestion enzyme trypsin. X! Tandem was set up to search a reverse concatenated subset of the Uniprot-human database (unknown version, 54442 entries) (only "Mudpit_GRP75_1: GRP75_1") also assuming trypsin. Sequest and X! Tandem were searched with a fragment ion mass tolerance of 0.60 Da and a parent ion tolerance of 10.0 PPM. Carbamidomethyl of cysteine was specified in Sequest and X! Tandem as a fixed modification. Met-loss of methionine, glu->pyro-Glu of the n-terminus, ammonia-loss of the n-terminus, gln->pyro-Glu of the n-terminus, deamidated of asparagine, oxidation of methionine, acetyl of lysine and the n-terminus and phospho of serine were specified in X! Tandem as variable modifications. Met-loss of methionine, met-loss+Acetyl of methionine, deamidated of asparagine, oxidation of methionine, acetyl of lysine and the n-terminus, phospho of serine and GG of lysine were specified in Sequest as variable modifications.

CRITERIA FOR PROTEIN IDENTIFICATION- Scaffold (version Scaffold_4.11.0, Proteome Software Inc., Portland, OR) was used to validate MS/MS based peptide and protein identifications. Peptide identifications were accepted if they could be established at >95.0% probability. Peptide Probabilities from X! Tandem were assigned by the Scaffold Local FDR algorithm. Peptide Probabilities from Sequest were assigned by the Peptide Prophet algorithm[67] with Scaffold delta-mass correction. Protein identifications were accepted if they could be

established at >99.0% probability and contained at least two identified peptides. Protein probabilities were assigned by the Protein Prophet algorithm[68]. Proteins that contained similar peptides, and could not be differentiated based on MS/MS analysis alone were grouped to satisfy the principles of parsimony.

## Subcellular Ca²⁺ measurements

AML12 cells stably expressing mitochondrial targeted ratiometric calcium sensor, 4mitD3-CPV[69] or primary hepatocyte were plated in a black, clear bottom 96 well plate (3603; Costar), and transduced with 4mitD3-CPV adenovirus for 24 h. The cells are then washed, and replaced the medium with Ca²⁺-free Krebs-Ringer bicarbonate (KRB) buffer. Fluorescence signals were monitored using two emission filters, YFP (540 nm) and CFP (490 nm) by ImageXpress Micro confocal (Molecular Devices). Fluorescence images was recorded every 10 s, and 90 s later, 100 μM (final concentration) ATP was injected. YFP and CFP fluorescence ratio were quantified using MetaXpress software (Molecular devices). 4mitD3-CPV plasmid used for stable cell line preparation, and adenoviral construct was kindly provided by Dr. Kyu-Sang Park (Yonsei University Wonju College of Medicine, Wonju, South Korea). Cytosolic Ca²⁺ levels were measured using the Fura-2 QBT™ Calcium Kit (#R6139; Molecular Devices) as described in the manufacturer's protocol. Briefly, cells were loaded with Fura-2 for 1 h and excited at 340 nm and 380 nm, with emission at 510 nm using FlexStation3 (Molecular Devices). Fluorescence signals were measured every 10 s and 60 s later, 100 μM ATP was injected.

## Mitochondrial oxygen consumption rate (OCR) measurement

Cells were seeded in Seahorse XF96 plate overnight, and treated with/without 100 mM EtOH for 24 h. Before OCR measurement, cells were washed and equilibrated for 1 h at 37 °C in XF base medium (102353-100; Agilent technologies) supplemented with 1X GlutaMAX (35050; Gibco), 1 mM sodium pyruvate (S8636; Sigma-Aldrich), and 25 mM glucose (G7528; Sigma-Aldrich) (pH 7.4). Pre-hydrated sensor cartridge was loaded with mitochondrial inhibitors to deliver a final concentration of 1 μM oligomycin (75351; Sigma-Aldrich), 1 μM FCCP (C2920; Sigma-Aldrich), and 0.5 μM rotenone (R8875; Sigma-Aldrich) with 0.5 μM Antimycin A (A8674; Sigma-Aldrich), and measured using Seahorse XFe96 analyzer (Agilent technologies). OCR values were normalized with protein concentration.

## Lipid-TOX staining

Cells were seeded in a black, clear bottom 96 well plate (3603; Costar). At the end of the experiment, cells were pre-stained with NucBlue (R37605; Thermofisher Scientific), washed (PBS), fixed (4% paraformaldehyde), and stained with LipidTOX neutral lipid stain (H34475; Life technologies corporation). Images were captured, and analyzed using ImageXpress Micro confocal and MetaXpress software (Molecular devices), respectively.

## Mitochondrial ROS and membrane potential measurement

Cells were plated in a black, clear bottom 96 well plate (3603; Costar). At the end of the experiment, cells were loaded with MitoSOX (M36008; Invitrogen) fluorescent dye to detect mitochondria superoxide or JC-1 (T3168; Invitrogen) fluorescent dye to detect mitochondrial membrane potential (MMP) for 10 min and 30 min, respectively, as described in the manufacturer's instruction. The fluorescence intensity was measured using FlexStation3 (Molecular Devices) and the mean fluorescence intensity (MFI) was normalized with the total nuclei (NucBlue stained) count per well.

## Mitochondrial DNA content

Genomic DNA (gDNA) and mitochondrial DNA (mtDNA) were extracted from the liver tissues using DNeasy Kit (69504; Qiagen) as previously described[70]. Briefly, 2.5 ng DNA were used to quantify the mtDNA and normalized with the gDNA using mitochondrial ND1 primers (Forward: CCTATCACCCTTGCCATCAT; Reverse: GAGGCTGTTGCTTGTGTGAC) and nuclear Pecam1 primers (Forward: ATGGAAAGCCTGCCATCATG; Reverse: TCCTTGTTGTTCAGCATCAC) by qPCR.

## Biochemical analysis

Serum aspartate transaminase (AST) and alanine transaminase (ALT) levels were measured with an automatic biochemistry analyzer 7020 (Hitachi). Triglyceride (TG) content in the liver homogenate was measured using the TG Quantification Colorimetric Kit (K622-100; BioVision) by following the manufacturer's instruction, and measured using micro-well spectrophotometer (Molecular Devices).

## Histological analysis

For immunohistochemistry, liver tissues were fixed with 4% PFA, embedded in paraffin, and immunohistochemistry analysis was performed. In short, after heat-induced epitope retrieval, paraffin-embedded sections were incubated in 3% H₂O₂, and blocked in 3% normal serum buffer. Sections were incubated with PDK4 antibody (NBP1-54723; Novus) overnight at 4 °C. Vectastain Elite ABC Staining Kit and DAB Peroxidase Substrate Kit (Vector Laboratories, Burlingame, CA) were used to visualize the staining. Hematoxylin and eosin (H&E) staining were performed according to the standard procedures. For Oil Red O staining, cryosections of OCT-embedded liver tissue samples were fixed in 10% formalin, stained with Oil Red O, washed with 60% isopropanol, and analyzed using Olympus Slideview VS200.

## Statistics and reproducibility

Graphs were plotted, and statistical significance was analyzed using GraphPad Prism 8 (GraphPad Software, La Jolla, CA). Images presented in the figures are the representatives of three or more independent experiments/samples with similar results. Statistical significance analysis was performed by two-tailed unpaired $t$-test for comparisons between two groups and one-way or two-way analysis of variance (ANOVA) to compare more than two groups applied with Dunnett's or Tukey's multiple comparison test from three or more independent experiments as specified in each figure legend. Statistical analysis was considered significant at $p$-value < 0.05 and it is indicated by the following annotations: *$p < 0.05$, **$p < 0.01$, ***$p < 0.001$, and ****$p < 0.0001$. The error bars are presented as mean ± standard mean error (SEM).

## Reporting summary

Further information on research design is available in the Nature Portfolio Reporting Summary linked to this article.

## Data availability

The mass spectrometry data have been deposited to the ProteomeXchange Consortium via the PRIDE[67] partner repository with the dataset identifier PXD039478. All other supporting data are included in the paper and supplementary information file. Source data are provided with this paper.

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

## Acknowledgements

We thank the team at the Department of Surgery at John Hopkins Hospital under the funding from the NIAAA (R24AA025017 clinical resources) for providing human liver tissue samples from patients with ALD. This work was supported by the National Research Foundation of Korea (NRF) grants, funded by the Ministry of Science and ICT (MSIT), Republic of Korea—2017R1A2B3006406, 2022R1A2B5B03001929 and the Korea Health technology R&D Project through the Korea Health Industry Development Institute (KHIDI) grants, funded by the Ministry of Health & Welfare, Republic of Korea—HI16C1501 and HR22C1832 to I.-K.L.; Grant funded by Ministry of Health & Welfare, Republic of Korea—HR22C1832 and NRF grant—2021R1A5A2021614 to J.-H.J.; KBRI basic research program through Korea Brain Research Institute grant, funded by the MSIT—23-BR-01-03 to J.Y.M. NRF grant—2021R1F1A1061393 to D.C.; NRF grants—2019R1I1A1A01062408 and 2022R1A2C1007857 to T.T.; National Institute of Health (NIH) grants, K01AA26385 to Z.Y; R01 AA025208, R01AA030312, U01 AA026917, UH2/3 AA026903, VA Merit Award 1I01CX000361, Dean's Scholar from Indiana University School of Medicine to S.L.

## Author contributions

T.T., S.L., and I.-K.L. designed the project. T.T., D.C., J.H.J., S.L., and I.-K.L. acquired the funding. T.T., D.C., B.-Y.P., I.S.S., N.H., P.K, M.K., J.M, ZY., and W.H.K. performed the in vitro and in vivo experiments. H.-J.K., Y.H.H., M.-K.J., J.Y.M., C.A.M., and J.Y.L. performed TEM imaging and analysis. B.-G.K. performed mass-spectrometric analysis. T.T. and D.C. designed and generated all the plasmids. T.T., M.K., H.L., J.Y.L, C.W.L., and D.C. performed imaging, mitochondrial function, qPCR, biochemical assay, histological analysis, and molecular/cellular techniques. R.S provided the human liver tissue samples. T.T., D.C., R.A.H., M.-J.K., J.H.J., S.L., and I.-K.L. analyzed and interpreted the experimental data. S.L. and I.-K.L. supervised the study. T.T., S.L., and I.-K.L. wrote the manuscript. D.C., M.-J.K., N.J.S., R.A.H., J.H.J., S.L., and I.-K.L. reviewed and edited the manuscript.

## Competing interests

The authors declare no competing interests.

## Additional information

[1]Research Institute of Aging and Metabolism, Kyungpook National University, Daegu, Republic of Korea. [2]Leading-Edge Research Center for Drug Discovery and Development for Diabetes and Metabolic Disease, Kyungpook National University Hospital, Daegu, Republic of Korea. [3]Neural Circuit Research Group, Korea Brain Research Institute, Daegu, Republic of Korea. [4]Bio-Medical Research Institute, Kyungpook National University Hospital, Daegu, Republic of Korea. [5]Center for Genomic Integrity, Institute for Basic Science (IBS), Ulsan, Republic of Korea. [6]Electron Microscopy Research Center, Korea Basic Science Institute, Ochang, Chungbuk, Republic of Korea. [7]Department of Medicine, Daegu Catholic University, Daegu, Republic of Korea. [8]Electron Microscopy Core, Indiana University School of Medicine, Indianapolis, IN, USA. [9]Department of Pathology and Laboratory Medicine, Indiana University School of Medicine, Indianapolis, IN, USA. [10]Department of Pathology and Laboratory Medicine, Emory University School of Medicine, Atlanta, GA, USA. [11]Department of Surgery, Louisiana State University Health Science Center, New Orleans, LA, USA. [12]Division of Gastroenterology and Hepatology, Department of Medicine, Indiana University School of Medicine, Indianapolis, IN, USA. [13]Department of Internal Medicine, Kyungpook National University Chilgok Hospital, Daegu, Republic of Korea. [14]Department of Biochemistry and Molecular Biology, Indiana University School of Medicine, Indianapolis, IN, USA. [15]Department of Internal Medicine, School of Medicine, Kyungpook National University, Kyungpook National University Chilgok Hospital, Daegu, Republic of Korea. [16]Richard L. Roudebush VA Medical Center, Indianapolis, IN, USA. [17]Department of Internal Medicine, School of Medicine, Kyungpook National University, Kyungpook National University Hospital, Daegu, Republic of Korea. [18]These authors jointly supervised this work: Suthat Liangpunsakul, In-Kyu Lee. ✉e-mail: sliangpu@iu.edu; leei@knu.ac.kr

