## [Peer Review File · Nature Communications]

Enhanced Ca²⁺-channeling complex formation at the ER-mitochondria interface underlies the pathogenesis of alcohol-associated liver diseaseREVIEWER COMMENTS

Reviewer #1 (Remarks to the Author):

Alcohol-associated liver disease (ALD) is associated with enhanced liver expression of PDK4 which is known to induce metabolic shift and impact several signaling pathways. Moreover, PDK4 has been shown to mediate lipogenesis and to contribute to the pathogenesis of nonalcoholic steatohepatitis. Thoudam et al. demonstrate that PDK4 phosphorylates the GRP75 chaperone at three Ser/Thr sites (T120, S266, T267), thereby promoting the formation of the IP3R1-GRP75-VDAC1 complex (MCC) and inducing mitochondrial Ca²⁺ uptake. Increased liver expression of PDK4 in ALD is associated with elevated formation of MCC and mitochondria-associated ER membranes (MAM), while PDK4 deficiency suppresses MCC and MAM formation and prevents ALD. Finally, mimicking the impact of PDK4 using an ER-mitochondrial linker in ethanol-fed Pdk4^{-/-} mice reproduces the ALD liver phenotype, thus supporting that PDK4-induced MCC formation contributes to ALD pathogenesis. The manuscript is clear and sound, but some points need to be clarified.

The authors focus on IP3R1 for the complex with GRP75 and VDAC1, but the latter interacts with other IP3R isoforms. Is the association with IP3R1 specific to ALD? Is this interaction altered by GRP75 phosphorylation status?

Measurements of mitochondrial functions (MMP, ROS, oxygraphy) are performed on whole cells in a steady state and do not take into account the fact that PDK4 can alter mitochondrial density and protein expression and activity of OXPHOS complexes. PDK4 has also been reported to alter expression of antioxidant enzymes. The manuscript should be clarified to avoid misinterpretation of the results as a whole, e.g. increased mitochondrial Ca²⁺ uptake induces decreased OXPHOS activity (outside a pro-apoptotic situation) as well as decreased MMP is associated with increased mitochondrial Ca²⁺ uptake and ROS production.

Statistics : The one-way ANOVA was used extensively, whereas the experimental design required a two-way ANOVA (Fig. 5 to 7, Suppl Fig. 3 to 5). Please justify and correct when necessary.

Minor comments:

- Page 3, line 9 : Change the format of the references
- Page 3, lines 26-27 : The sentence should be revised
- Page 3, line 28-30 : Add the references
- Page 3, introduction : It would be useful to have a paragraph summarising the hypothesis and the objectives of the work at the very end.
- Page 5, line 5 : Correct the statement associated with reference 31: Incomplete oxidation of fatty acids and accumulation of derived metabolites are responsible for enhanced mitochondrial Ca²⁺ uptake, not vice versa.
- Page 7 : We are confronted with a gap in understanding the association between mitochondrial dysfunction and intracellular lipid accumulation. It would be interesting to explore fatty acid beta-oxidation and markers of mitochondrial and endoplasmic reticulum stress. Are the latter parameters reversed with the ER-mitochondrial linker ?
- Figure Suppl. 3A : Please provide the original WB for PDK4

Reviewer #2 (Remarks to the Author):

The paper examines the role of PDK4, GRP75, and MAMs in alcoholic liver disease (ALD). The discovery that PDK4, GRP75, and MAMs play a role in ALD is noteworthy. While there are some very interesting findings, there are also some concerns.

Major points

- 1) Alcohol does not effect hepatocytes in the liver homogeneously, rather alcohol-induced injury preferentially affects perivenous hepatocytes versus periportal hepatocytes (Guerra 1987). It is therefore very likely that mitochondrial alterations are going to be very different in different hepatocyte populations. Figure 1 exams mitochondrial alterations such as area, MAM, and ER by examining 20-30 data points. This is an extremely low number to characterize mitochondria considering that different hepatocyte populations may have different mitochondrial changes. How can the authors be certain that both periportal and perivenous hepatocytes were considered or that there was not a bias towards one population? The number of mitochondria counted and the exact number of hepatocytes are not clear (except maybe 1l), but seem very inadequate. Figure 1 should be redone considering measurements of a high number of mitochondria in a high number random number of hepatocytes, which should include both periportal and perivenous hepatocytes. If you could distinguish mitochondrial changes in periportal and perivenous hepatocytes, the study would be even better.
- 2) Primarily hepatocytes de-differentiate over time, including changes in mitochondrial respiration. Thus experiments involving primary hepatocytes are problematic (Fig 5) especially when experiments go for 24 hours where significant de-differentiation and loss of mitochondrial respiration likely occurred. Did the authors confirm that mitochondrial responses following 24 hours were very similar to mitochondrial responses in freshly isolated hepatocytes? In addition, Figure 5 incubates 100 mM alcohol, which is an acute alcohol model, which is very different from the chronic model feeding studies in many other figures. The authors need to recognize that acute alcohol treatment and chronic alcohol feeding are different models and discuss the implications.
- 3) All cells treatment were performed with 100 mM alcohol, which is on the high side of dosing. On the other hand, alcohol evaporates very rapidly in culture media at 37 degrees. Thus the actual dose given and incubation time (24 hours) and not accurate. The authors should consider either using method stop prevent alcohol evaporation or a rough measure of actual alcohol delivered to truly understand this acute model. Several other doses of alcohol would also be helpful for some measurements.
- 4) Most of the Western blots lack densometry, thus the extent of change is difficult to determine. There should be densometry included in the figures for the Westerns.
- 5) The notion that ALD or NAFLD only involve mitochondrial dysfunction is a very narrow view of mitochondrial alterations that occur with liver disease. Many works have suggests that mitochondrial respiration increases with ALD depending on model (Han 2017) and that mitochondrial alterations are very dynamic in liver diseases (Shum 2020). Thus the mitochondrial alterations observed in the paper

may not be purely pathological, but part of an adaptive process, that contributes to liver disease. Some type of discussion regarding dynamics and adaptation would provide a more broad perspective.

6) Anastasia 2021 has suggested that all mitochondria in the liver were surrounded by the ER. The findings of this paper seem to differ. Some discussion of the difference would be helpful.

7) The experiments with the linkers (Fig 7) are interesting. This leads to the question, can linkers alone cause steatosis or does some sort of metabolic stress like alcohol need to be included to cause liver disease. Can prolonged treatment with linkers lead to steatosis similar to chronic alcohol feeding?

Reviewer #3 (Remarks to the Author):

In this study, Thoudam et al. display that mitochondria-associated membranes (MAM) are increased during alcohol-associated liver diseases (ALD). Mechanistically, they point out the role of PDK4 in elevated MAM-formation induced by ethanol-diet, and in subsequent mitochondrial calcium elevation and associated mitochondrial dysfunction. Molecularly, they demonstrate that increased MAM formation relies on GRP75 phosphorylation by PDK4.

This work is of general interest in the field and has potentially broad relevance. Nevertheless, some of the presented results, particularly related to calcium imaging (Point 2), are not fully supported by data, and additional experiments and evidence could improve the overall quality of this important work.

- Major points

Technical aspects

1- On PLA experiments: Fig 2J. Fig 3I. Fig 4B. Fig 5A

The PLA experiments coupling ITPR1 and VDAC1 antibodies in Fig 2J display some background, that may be due to tissue processing. In order to avoid any misinterpretation in other PLA experiments and as it is known to be a sensitive tool, I would suggest to the authors to display in parallel 1°) one PLA using only ITPR1 or VDAC1 antibodies alone, and 2°) one PLA ITPR1-VDAC1 on AML12 cells using knocking down ITPR1 or VDAC1 to ensure a global quality control of the PLA.

2- On calcium imaging: Fig 3K-L-M, Fig 4D-E-F, Fig 5G.

2.1- For basal resting mitochondrial calcium level. Rhod2-AM is a non-ratiometric calcium probe and may then depend to the capacity of each cell to capture and retain the probe. The authors can only conclude for stimulated mitochondrial calcium levels and but not for the basal resting mitochondrial calcium level. In order to evaluate this last one, the authors should use a ratiometric probe (10.1038/ncomms5153) as there has been multiple developed for almost ten years in the field. This crucial tool will allow the normalization to the probe itself by different emission according to calcium binding state of the probe.

2.2- For histamine-stimulated mitochondrial calcium level. Even the results look coherent and as the

authors use a non-ratiometric calcium probe, the fluorescence increased upon histamine stimulation should be normalized to the resting levels and the fold ratio then considered. The peak (delta) and/or the area under the curve for each condition should be quantified and displayed by the authors to confirm an enhanced sensitivity to histamine with EtOH, reversed by the different GRP75 mutant overexpressions. More importantly, the pattern of mitochondrial calcium uptake upon histamine injection is totally different between Fig 3K and Fig 4D. In Fig 4D, there is not the typical progressive decrease of mitochondrial calcium (observed in Fig 3K for instance), pointing out technical issue in this specific experiment.

Histamine stimulates ITPRs calcium channel (IP3R receptors) through IP3 generation. Mitochondrial calcium uptake may depend also on the ability of ER to release calcium. As ITPR1 expression is different in various conditions (Supp Table 1, Fig 4A), it could particularly impact ER-calcium release also. In order to exclude the potential increase of ER calcium release in PDK4-overexpressed or EtOH-treated cells. I strongly suggest to the authors to evaluate ER calcium release or cytosolic increase upon histamine, also using a ratiometric probe (10.1038/ncomms5153) (Fura-2 for cytosolic one).

Theoretical aspects

3- An interesting and not raised point from this study is how EtOH does regulate PDK4 expression in liver. Do we know transcriptional factors regulating PDK4? Do we know EtOH-induced transcriptional factors? I encourage the authors of the study to discuss these questions.

4- As suggesting by the authors in the discussion, the use of a PDK4 inhibitor (or at least Pan-PDKs inhibitors) to confirm rescue of some of the main EtOH-mediated effects (such as MAM formation, GRP75 phosphorylation, mitochondrial calcium uptake, ROS generation or mitochondrial functions) would be of interest and would further reinforce the consideration of targeting PDK4 during alcoholic-liver disease.

- Minor points

5- Fig 1B: It appears that there is an increased mitochondria number in the liver of EtOH-treated mice. The authors quantified MT area in Fig 1C, but a better indicator would have been the length of mitochondria (i.e. perimeters), as this is precisely this surface which is in contact with ER. Same remark for the ER length, which looks like more dispersed and bigger in liver of EtOH-treated mice. As ER surfaces not reflect the truth of the quantification

6- Fig 1C-D-E-G: What does represent one dot in the graph? One cell? In the legends, the authors mentioned only n=3 independent mice. To strengthen this data, is it possible for the authors to precise both number of cells and also mitochondria evaluated approximately in each mice. Furthermore, if they pool the individual data, is it possible to provide a test showing that no differences between mice from the same group?

7- Supp Table 1 displays an increase of mRNA levels of Hspa9 gene, which encodes for GRP75 (+0.361

log₂FC), but also of *Itpr1* gene (+0.573 log₂FC). How the authors can not exclude that EtOH-Mam formation is not the result of this increased expression, independently of GRP75?

8- Fig 2F. The authors could add a relative quantification of PDK4 protein level as it does not appear to be upregulated according to the blot.

9- Page 3 - Line 22: The authors mention the aberrant increase of MAM and subsequent mitochondrial dysfunction during hepatic insulin resistance. Because of the role of MAM in multiple cellular phenotypes related to hepatic insulin resistance and/or liver functions, autophagy (DOI: 10.3389/fcell.2020.00595), inflammation (DOI: 10.1038/s41419-017-0027-2), apoptosis (DOI: 10.1126/science.1189157) and also cellular senescence (DOI: 10.1038/s41467-021-20993-z). The authors should discuss the consequence of the aberrant induction of MAM on cellular phenotype and how it relates to the liver phenotype they observed.

10- Page 3 - Line 26.27: The sentence “Whereas, ...high fat diet-induced steatosis” should be corrected.

11- Fig5: The main known function of PDK4 is to phosphorylate and inhibits PDH, reducing mitochondrial Acetyl-CoA generation and subsequent oxidative phosphorylation. Manipulating PDK4 has been shown to modify fatty acid oxidation, but also glycolysis (DOI: 10.1038/s41598-017-09163-8, DOI:10.1016/j.mito.2019.07.009). The authors should discuss that point and tone down the conclusions based on role of PDK4 only in MAM formation.

12- The link between the aberrant enhanced MAM/mitochondrial dysfunction and the pathological phenotype of the liver observed in EtOH-fed mice remains unclear, especially related to lipid accumulation. Can the authors comment on this link between MAM and lipid droplets, related to lipogenesis/lipolysis.

13- Finally, beyond AST (which is not specific for liver damage, and which is the result also of other tissue damages, such as muscle), ALT and triglycerides levels, the proper liver functions have not been clearly evaluated in ED-fed mice. Do ED-fed mice display hyperglycemia at basal level? Do they present the inability of glucose clearance or some insulin resistance? Could it be possible to evaluate (or at least discuss) the function of these livers? And also reusing it in the some of the different presented conditions?

Reviewer #1 (Remarks to the Author):

Alcohol-associated liver disease (ALD) is associated with enhanced liver expression of PDK4 which is known to induce metabolic shift and impact several signaling pathways. Moreover, PDK4 has been shown to mediate lipogenesis and to contribute to the pathogenesis of nonalcoholic steatohepatitis. Thoudam et al. demonstrate that PDK4 phosphorylates the GRP75 chaperone at three Ser/Thr sites (T120, S266, T267), thereby promoting the formation of the IP3R1-GRP75-VDAC1 complex (MCC) and inducing mitochondrial Ca²⁺ uptake. Increased liver expression of PDK4 in ALD is associated with elevated formation of MCC and mitochondria-associated ER membranes (MAM), while PDK4 deficiency suppresses MCC and MAM formation and prevents ALD. Finally, mimicking the impact of PDK4 using an ER-mitochondrial linker in ethanol-fed Pdk4^{-/-} mice reproduces the ALD liver phenotype, thus supporting that PDK4-induced MCC formation contributes to ALD pathogenesis. The manuscript is clear and sound, but some points need to be clarified.

- We would like to thank the reviewer for the insightful critiques/comments and an overall positive outlook towards our manuscript. To address the reviewer's suggestions, we have performed additional experiments. We hope to have addressed the reviewer's concerns satisfactorily. The revisions in the manuscript are written in red for your recognition. Please find below our point-by-point response to the comments (written in blue text).

The authors focus on IP3R1 for the complex with GRP75 and VDAC1, but the latter interacts with other IP3R isoforms. Is the association with IP3R1 specific to ALD? Is this interaction altered by GRP75 phosphorylation status?

- We agree with the reviewer's comment that there are three isoforms of IP3R. In the liver, IP3R1 and IP3R2 are predominantly expressed (Wojcikiewicz RJ. J Biol Chem. 1995). Here, we have looked at the effect of alcohol on IP3R1 and IP3R2 interaction with GRP75 in CD-fed vs ED-fed livers. We observed that IP3R1-GRP75 interaction was increased, whereas, IP3R2-GRP75 interaction was slightly decreased in ED-fed mice compared to CD-fed mice (added the data in Fig. 2K). Consistently, we also observed that EtOH treatment induced GRP75 interaction with IP3R1 but not with IP3R2 (attached below). These results suggest that IP3R1 is the predominant isoform involved in MCC complex formation in ALD. Additionally, earlier studies have reported that IP3R1 plays a more direct role than IP3R2 in generating mitochondrial calcium signals in hepatocytes and aberrant IP3R1 signaling was found to promote fatty liver disease (Feriod CN. et al., Hepatol Commun. 2017; Feriod CN et al., Physiol Endocrinol Metab. 2014). Therefore, in this study, we focused on IP3R1 isoform to evaluate MCC complex formation. However, the effect of GRP75 phosphorylation on MCC complex formation in cells or tissues where IP3R2/3 is the dominant isoform cannot be ruled out from our current conclusions.

Measurements of mitochondrial functions (MMP, ROS, oxygraphy) are performed on whole cells in a steady state and do not take into account the fact that PDK4 can alter mitochondrial density and protein expression and activity of OXPHOS complexes. PDK4 has also been reported to alter expression of antioxidant enzymes. The manuscript should be clarified to avoid misinterpretation of the results as a whole, e.g. increased mitochondrial Ca²⁺ uptake induces decreased OXPHOS activity (outside a pro-apoptotic situation) as well as decreased MMP is associated with increased mitochondrial Ca²⁺ uptake and ROS production.

- We have evaluated the effect of PDK4 deficiency on mitochondrial density, OXPHOS complexes and antioxidant protein expressions to confirm that the changes in mitochondrial function were not the outcome of altered mitochondrial OXPHOS and antioxidant enzymes in PDK4 deficient cells. However, no significant difference in mitochondrial DNA content (Supplementary figure 7A) and mitochondrial OXPHOS protein expression was observed in EtOH-treated hepatocytes or ED-fed mice liver compared to controls, except for Complex IV [cytochrome c oxidase subunit I (MT-CO1)] which was slightly higher in ED-fed *Pdk4*^{+/+} mice but not in ED-fed *Pdk4*^{-/-} mice (S Supplementary figure 5D, 7B-C). Previously, MT-CO1 overexpression was shown to promote ROS formation (DOI: 10.1016/j.mito.2019.07.002). Notably, among the antioxidant enzymes, glutathione peroxidases 4 (GPX4), an enzyme that protects against cellular lipid peroxidation was significantly reduced in ED-fed *Pdk4*^{+/+} mice and in hepatocytes treated with EtOH compared to controls but, this was restored in both ED-fed *Pdk4*^{-/-} mice and EtOH-treated *Pdk4*^{-/-} hepatocytes (Supplementary figure 5D and Supplementary figure 7B, 7D-F). This indicates that PDK4 deficiency may prevent alcohol-induced GXP4 depletion. Together, these results suggest that alcohol-induced mitochondrial Ca²⁺ uptake has a profound effect on OXPHOS activity, MMP and mitochondrial ROS production.

Statistics : The one-way ANOVA was used extensively, whereas the experimental design required a two-way ANOVA (Fig. 5 to 7, Suppl Fig. 3 to 5). Please justify and correct when necessary.

- Thank you for pointing out this mistake. We have used two-way ANOVA for the analysis. We have corrected this mistake in the revised manuscript.

Minor comments:

- Page 3, line 9 : Change the format of the references

- We have formatted the references.

- Page 3, lines 26-27 : The sentence should be revised

- We have revised the sentence.

- Page 3, line 28-30 : Add the references

- The references in line 33 also correspond to the line 28-30. To avoid confusion, we have revised the sentences and the reference has be added.

- Page 3, introduction: It would be useful to have a paragraph summarising the hypothesis and the objectives of the work at the very end.

- We have added the paragraph summarizing the hypothesis and the objectives of the work at the very end of the introduction section.

- Page 5, line 5 : Correct the statement associated with reference 31: Incomplete oxidation of fatty acids and accumulation of derived metabolites are responsible for enhanced mitochondrial Ca²⁺ uptake, not vice versa.

- We have corrected the statement in the revised manuscript.

- Page 7 : We are confronted with a gap in understanding the association between mitochondrial dysfunction and intracellular lipid accumulation. It would be interesting to explore fatty acid beta-oxidation and markers of mitochondrial and endoplasmic reticulum stress. Are the latter parameters reversed with the ER-mitochondrial linker?

- It is an interesting point raised by the reviewer. The fundamental function of MAM includes phospholipid synthesis besides its critical role in regulating Ca²⁺ transport (Flis VV, Daum G. et al Spring Harb Perspect Biol. 2013). Recently several interesting reports have revealed a completely new angle suggesting lipid droplet formation at MAM. Multiple proteins involved in lipid droplet biosynthesis were found to be localized at MAM such as (mitogurdin2) MIGA2 (Freyre CAC et al Mol cell, 2019; Kim H. et al. Nat Comm, 2022), Seipin (Combot Y et al. Cell rep, 2021), ORP5 and ORP8 (Guyard V et al. J Cell Biol, 2022). Importantly, the integrity of ER-mitochondria contact sites is crucial for orchestrating LD biogenesis and maturation (Guyard V et al. J Cell Biol, 2022).

Interestingly, we did not observe any significant difference in lipid accumulation or gross morphologic difference and TG content between mock control and linker-expressing mice liver, which is consistent with the previous report that linker expression alone does not produce gross morphological changes but exacerbated ER stress in diet-induced obese mice liver (Arruda AP et al, Nat Med 2014). We have previously reported that genetic ablation of PDK4 failed to reversed linker-induced JNK1 activation and insulin resistance in skeletal muscle (Thoudam et al Diabetes 2019).

- Figure Suppl. 3A : Please provide the original WB for PDK4

- Here is the original WB image for PDK4 in Figure Suppl. 3A.

Reviewer #2 (Remarks to the Author):

The paper examines the role of PDK4, GRP75, and MAMs in alcoholic liver disease (ALD). The discovery that PDK4, GRP75, and MAMs play a role in ALD is noteworthy. While there are some very interesting findings, there are also some concerns.

- We would like to thank the reviewer for the in-depth review and highly appreciate the reviewer's valuable comments. We have thoroughly studied the comments and we have put our best effort to address all the concerns. We hope that the reviewer finds our additional data and following clarification satisfactory. The revisions in the manuscript are written in red for your recognition. Please find below our point-by-point response to the comments (written in blue text).

Major points

1) Alcohol does not effect hepatocytes in the liver homogeneously, rather alcohol-induced injury preferentially affects perivenous hepatocytes versus periportal hepatocytes (Guerri 1987). It is therefore very likely that mitochondrial alterations are going to be very different in different hepatocyte populations. Figure 1 exams mitochondrial alterations such as area, MAM, and ER by examining 20-30 data points. This is an extremely low number to characterize mitochondria considering that different hepatocyte populations may have different mitochondrial changes.

How can the authors be certain that both periportal and perivenous hepatocytes were considered or that there was not a bias towards one population?

The number of mitochondria counted and the exact number of hepatocytes are not clear (except maybe 1I), but seem very inadequate. Figure 1 should be redone considering measurements of a high number of mitochondria in a high number random number of hepatocytes, which should include both periportal and perivenous hepatocytes. If you could distinguish mitochondrial changes in periportal and perivenous hepatocytes, the study would be even better.

- Reviewer pointed out a very crucial point that alcohol does not affect hepatocytes in the liver homogeneously, rather alcohol-injury preferentially affects perivenous hepatocytes compared to periportal hepatocytes. We were aware of this issue from the beginning of the study, and we have looked at the perivenous region of the liver. Now, we added the description of the region analyzed in the manuscript for the EM figures.

To address this specific concern of the reviewer, we have applied the same parameters (ER perimeter, mitochondria perimeter, % of mitochondria in apposition to ER and MAM length) and analyzed the periportal region. As the reviewer pointed out, we found some interesting differences between two regions in the ED-fed mice livers -

1. Mitochondrial perimeter was decreased in the perivenous region (Figure 1B-C) but increased mitochondrial perimeter in the periportal (morphologically more round and blotted mitochondria) (Supplementary figure 1A-B) was observed in the ED-fed mice.

2. No significant changes were observed in the ER perimeter between CD and ED-fed mice in the perivenous region (Figure 1A & 1C), meanwhile, the ER perimeter was significantly decreased in the periportal region (Supplementary figure 1A & 1C) of the ED-fed mice compared with the CD-fed mice.

3. Interestingly, the percentage of mitochondria in close apposition to the ER was increased in both the perivenous region (Figure 1E) and the periportal region (Supplementary figure 1D). However, significant increase in the MAM length was observed in the perivenous region (Figure 1G) but not in the periportal region (Supplementary figure 1F).

Together, these results revealed a differential effect of alcohol on mitochondria and ER ultrastructure in the pericentral and periportal hepatocytes population. Notably, despite having difference in the MAM length, the association between ER and mitochondria was enhanced in both regions, suggesting alcohol promotes MAM formation but more prominently in the pericentral region.

Regarding the no. of points presented in the graph, the dot/points represent the number of electron microscopic fields analyzed. We used number of microscopic fields as “n” number as we couldn’t plot graph using GraphPad prism software with “n” number more 256 which we have exceeded. To address this concern, we have added the actual number of mitochondria analyzed in the study in the corresponding figure (which is 260 mitochondria for CD and 390 mitochondria for ED in the perivenous region and 160 and 230 mitochondria for CD and ED, respectively, in the periportal region).

Lastly, we have replaced the ER and mitochondrial area with perimeter (Figure 1C-D & Supplementary figure 1B-C).

2) Primarily hepatocytes de-differentiate over time, including changes in mitochondrial respiration. Thus experiments involving primary hepatocytes are problematic (Fig 5) especially when experiments go for 24 hours where significant de-differentiation and loss of mitochondrial respiration likely occurred. Did the authors confirm that mitochondrial responses following 24 hours were very similar to mitochondrial responses in freshly isolated hepatocytes? In addition, Figure 5 incubates 100 mM alcohol, which is an acute alcohol model, which is very different from the chronic model feeding studies in many other figures. The authors need to recognize that acute alcohol treatment and chronic alcohol feeding are different models and discuss the implications.

- Taking the reviewer’s concern into consideration, we have performed an additional experiment to compare the oxygen consumption rate between hepatocytes that were freshly isolated and cultured for 24 hours and evaluated the possibility of a loss of mitochondrial respiration due to potential de-differentiation. Interestingly, freshly isolated hepatocytes had lower maximal respiration compared to the hepatocytes cultured for 24 hours but no significant difference was observed in the basal or the ATP-linked respiration. However, we believe that the difference in maximal OCR in this experiment may be due to limited time for adaptation in the culture environment for the freshly isolated hepatocytes (6 hours) compared to the hepatocytes kept in culture for 24 hours. Together, this result indicates that the 24-hour culture time point do not lead to reduction of mitochondrial respiration.

- In this study, we focused on the role of mitochondrial dysfunction in the pathogenesis of ALD. We observed that 100mM alcohol treatment for 24 hours was sufficient to promote MAM formation, mitochondrial Ca²⁺ accumulation, mitochondrial dysfunction, and lipid accumulation *in vitro*. Therefore, we used 24-hour time point throughout our study.

3) All cells treatment were performed with 100 mM alcohol, which is on the high side of dosing. On the other hand, alcohol evaporates very rapidly in culture media at 37 degrees. Thus the actual dose given and incubation time (24 hours) and not accurate. The authors should consider either using method stop prevent alcohol evaporation or a rough measure of actual alcohol delivered to truly understand this acute model. Several other doses of alcohol would also be helpful for some measurements.

- In this study, we have used a major AA component of the microsomal EtOH-oxidizing system, Cyp2E1 expression, as an indicator for EtOH exposure *in vitro* and *in vivo*. We found that 100mM EtOH treatment for 24h was the most effective dose to induce Cyp2E1 in both AML12 cells and primary hepatocytes (Supplementary figure 2A-2B). In the beginning, we tried both the stop prevent alcohol evaporation method using closed a chamber or refreshing EtOH-containing media every 12 hours to maintain the EtOH exposure to the cells. We found that both of the methods induced comparable levels of Cyp2E1. However, we choose the latter EtOH treatment method in this study as the stop prevent alcohol evaporation method was not possible to apply while handling multi-well plates.

We also have performed additional experiments to evaluate the dose-dependent effects of EtOH on MCC complex formation by *in situ* PLA. We observed an EtOH dose-dependent increase in MCC complex formation but statistical significance was observed only at 100mM EtOH dose (Supplementary figure 2G-2H) which is in line with the observation that 100mM EtOH dose significantly induced mitochondrial Ca²⁺ accumulation (Figure 5G and supplementary figure 9D), ROS generation (Figure 5H and supplementary figure 9E), lipid accumulation (supplementary figure 9F). Together, this result indicates that 100mM EtOH is the optimal dose to mimic ALD *in vitro*.

4) Most of the Western blots lack densitometry, thus the extent of change is difficult to determine. There should be densitometry included in the figures for the Westerns.

- We have added the densitometric analysis for all the western blot images.

5) The notion that ALD or NAFLD only involve mitochondrial dysfunction is a very narrow view of mitochondrial alterations that occur with liver disease. Many works have suggests that

mitochondrial respiration increases with ALD depending on model (Han 2017) and that mitochondrial alterations are very dynamic in liver diseases (Shum 2020). Thus the mitochondrial alterations observed in the paper may not be purely pathological, but part of an adaptive process, that contributes to liver disease. Some type of discussion regarding dynamics and adaptation would provide a more broad perspective.

It's a valid point raised by the reviewer as the variable effect of alcohol on mitochondrial function was observed in the earlier reports, Han et al (2017) observed that mitochondrial respiration was increased in alcohol-fed rats and in contrast, Gordon E R (J Biol Chem, 1973), Kayo Adachi et al (Acta Pathol Jpn., 1991) and Matsushashi T. et al. (Free Radical Biol & Med. 1998) found that ethanol decreases hepatic mitochondrial respiration in ALD rodent models. Furthermore, mitochondria in normal hepatocytes show relatively slow dynamics, which are very sensitive to suppression by ethanol exposure (Das S. et al Pflugers Arch - Eur J Physiol, 2012). We have discussed this issue in our revised version of the manuscript (please refer to the 3rd paragraph of the discussion section).

6) Anastasia 2021 has suggested that all mitochondria in the liver were surrounded by the ER. The findings of this paper seem to differ. Some discussion of the difference would be helpful.

In Anastasia et al, the method used to measure wrapER-mitochondria contact distance was 100nm and 10nm~30nm was considered as MAM. However, in our study we have considered <50nm as MAM as described (Giacomello M, Pellegrini L. Cell Death Differ. 2016.). We observed that 45% and 65% of the mitochondria are apposition with the ER (within 50nm distance) in perivenous (Figure 1E) and periportal region (Supplementary Figure 1D), respectively, in CD-fed mice liver. However, this was increased to 72% in the perivenous (Figure 1E) and 78% in the periportal region (Supplementary figure 1D) in ED-fed mice liver. Similarly, Anastasia et al also observed that the percentage of the mitochondrial perimeter covered by the wrapER and MAM was decreased in fed-state compared to fasted mice. Overall, these data indicate that although the association between wrapER and mitochondria is always present and extensive, the extent of the contact between these organelles is dynamically regulated by the metabolic changes that accompany the nutritional status of the animal. We have added the difference in the study in the discussion (please refer to the first paragraph of the discussion section).

7) The experiments with the linkers (Fig 7) are interesting. This leads to the question, can linkers alone cause steatosis or does some sort of metabolic stress like alcohol need to be included to cause liver disease. Can prolong treatment with linkers lead to steatosis similar to chronic alcohol feeding?

- We have evaluated the effect of linker alone on inducing liver steatosis by Oil Red O, H&E staining, and measuring triglyceride (TG) content. However, we did not observe any significant difference in lipid accumulation or gross morphologic difference and TG content between mock control and linker-expressing mice liver (Supplementary figure. 10C-10F). These findings are consistent with the previous report indicating that linker expression alone does not appear to produce any toxic effects, nor does it cause gross morphological changes within the liver or induce a metabolic phenotype (Arruda AP et al, Nat Med 2014). Together, these results suggest that induction of MAM alone in these current experimental settings does not induce hepatic steatosis.

Reviewer #3 (Remarks to the Author):

In this study, Thoudam et al. display that mitochondria-associated membranes (MAM) are increased during alcohol-associated liver diseases (ALD). Mechanistically, they point out the role of PDK4 in elevated MAM-formation induced by ethanol-diet, and in subsequent mitochondrial calcium elevation and associated mitochondrial dysfunction. Molecularly, they demonstrate that increased MAM formation relies on GRP75 phosphorylation by PDK4.

This work is of general interest in the field and has potentially broad relevance. Nevertheless, some of the presented results, particularly related to calcium imaging (Point 2), are not fully supported by data, and additional experiments and evidence could improve the overall quality of this important work.

- We would like to thank the reviewer for the insightful critiques/comments and an overall positive outlook towards our manuscript. We appreciate that the reviewer pointed out several important points that we missed and technique related concerns. We have put our best effort to address all the concerns. We hope that the reviewer finds our additional data and following clarification satisfactory. Please find below the point-by-point response to each of the individual comments. We hope that with additional suggested experiments and explanations we have been able to clarify all the concerns raised by the reviewer. The revisions in the manuscript are written in red for your recognition.

Please find below our point-by-point response to the comments (written in blue text).

- Major points

Technical aspects

1- On PLA experiments: Fig 2J. Fig 3I. Fig 4B. Fig 5A

The PLA experiments coupling ITPR1 and VDAC1 antibodies in Fig 2J display some background, that may be due to tissue processing. In order to avoid any misinterpretation in other PLA experiments and as it is known to be a sensitive tool, I would suggest to the authors to display in parallel 1°) one PLA using only ITPR1 or VDAC1 antibodies alone, and 2°) one PLA ITPR1-VDAC1 on AML12 cells using knocking down ITPR1 or VDAC1 to ensure a global quality control of the PLA.

We agree with the reviewer's comment that Fig 2J displays some background that may have been caused by the tissue processing because the background is distinguishable from the PLA blobs. We performed experiments to validate the antibody specificity used in the PLA by knocking down IP3R1 or VDAC1. Knockdown of IP3R1 or VDAC1 reduced the PLA and we found no cross-reactivity when only one antibody is used either IP3R1 or VDAC1. This data has been added in the Supplementary figure. 2C-2F to avoid any misinterpretation of other PLA experiments.

2- On calcium imaging: Fig 3K-L-M, Fig 4D-E-F, Fig 5G.

2.1- For basal resting mitochondrial calcium level. Rhod2-AM is a non-ratiometric calcium probe and may then depend to the capacity of each cell to capture and retain the probe. The authors can only conclude for stimulated mitochondrial calcium levels and but not for the basal resting mitochondrial calcium level. In order to evaluate this last one, the authors should use a ratiometric probe (10.1038/ncomms5153) as there has been multiple developed for almost ten years in the

field. This crucial tool will allow the normalization to the probe itself by different emission according to calcium binding state of the probe.

2.2- For histamine-stimulated mitochondrial calcium level. Even the results look coherent and as the authors use a non-ratiometric calcium probe, the fluorescence increased upon histamine stimulation should be normalized to the resting levels and the fold ratio then considered. The peak (delta) and/or the area under the curve for each condition should be quantified and displayed by the authors to confirm an enhanced sensitivity to histamine with EtOH, reversed by the different GRP75 mutant overexpressions. More importantly, the pattern of mitochondrial calcium uptake upon histamine injection is totally different between Fig 3K and Fig 4D. In Fig 4D, there is not the typical progressive decrease of mitochondrial calcium (observed in Fig 3K for instance), pointing out technical issue in this specific experiment.

- We have performed an additional experiment to reanalyze the mitochondrial Ca²⁺ flux using a mitochondria-target ratiometric calcium biosensor, 4MitD3-CPV (Palmer AE et al., Chem Biol. 2006). Detailed experimental description has been added in the materials and method section. We observed that PDK4 and EtOH-induced mitochondrial Ca²⁺ uptake was significantly reduced in GRP75 mutant expressing cells or by genetic ablation of PDK4 in hepatocytes (Figure 3K-M, Figure 4D-F, Figure 5G and supplementary figure 9D). We have replaced all the Rhod2 data with the 4MitD3-CPV data in the revised manuscript.

Histamine stimulates ITPRs calcium channel (IP3R receptors) through IP3 generation. Mitochondrial calcium uptake may depend also on the ability of ER to release calcium. As ITPR1 expression is different in various conditions (Supp Table 1, Fig 4A), it could particularly impact ER-calcium release also. In order to exclude the potential increase of ER calcium release in PDK4-overexpressed or EtOH-treated cells. I strongly suggest to the authors to evaluate ER calcium release or cytosolic increase upon histamine, also using a ratiometric probe (10.1038/ncomms5153) (Fura-2 for cytosolic one).

- We have performed an addition experiment to measure the cytosolic Ca²⁺ levels corresponding to Figure 3K and Figure 4D using Fura-2 ratiometric dye. However, we did not observe any defect in IP3R-mediated ER calcium release in GRP75 mutant expressing cells (Supplementary figure 4A & 4B).

Theoretical aspects

3- An interesting and not raised point from this study is how EtOH does regulate PDK4 expression in liver. Do we know transcriptional factors regulating PDK4? Do we know EtOH-induced transcriptional factors? I encourage the authors of the study to discuss these questions.

It is an interesting point that the reviewer has raised, Foxo1, PGC1a, ERR γ and C/EBP β are some of the known transcription factors that regulates *Pdk4* gene expression. Among those, C/EBP β , and ERR γ were shown to be highly induced in ALD, suggesting a potential involvement of these transcription factors in EtOH-mediated induction of PDK4 transcript level. We have added the information in the discussion section (please refer to the 2nd paragraph of the discussion section).

4- As suggesting by the authors in the discussion, the use of a PDK4 inhibitor (or at least Pan-PDKs inhibitors) to confirm rescue of some of the main EtOH-mediated effects (such as MAM formation, GRP75 phosphorylation, mitochondrial calcium uptake, ROS generation or

mitochondrial functions) would be of interest and would further reinforce the consideration of targeting PDK4 during alcoholic-liver disease.

We have performed additional experiments to address this interesting point raised by the reviewer. Dichloroacetic acid (DCA) has long been known as the most potent PDK inhibitor. In this study, we have examined the effect of DCA on EtOH-mediated effects (such as MAM formation, GRP75 phosphorylation, mitochondrial calcium uptake, and ROS generation) in AML12 cells. Interestingly, 2mM DCA treatment (co-treatment with EtOH) significantly suppresses EtOH-induced PDK4 expression, GRP75 phosphorylation, MAM formation, mitochondrial Ca²⁺ uptake, ROS generation and Lipid accumulation (Supplementary 9A-G). These results suggest that inhibition of PDK4 might be a viable therapeutic route to treat ALD.

- Minor points

5- Fig 1B: It appears that there is an increased mitochondria number in the liver of EtOH-treated mice. The authors quantified MT area in Fig 1C, but a better indicator would have been the length of mitochondria (i.e. perimeters), as this is precisely this surface which is in contact with ER. Same remark for the ER length, which looks like more dispersed and bigger in liver of EtOH-treated mice. As ER surfaces not reflect the truth of the quantification.

- We have replaced the ER and mitochondrial area with perimeter. In addition, we have additional data comparing the mitochondrial and ER perimeter in the perivenous region and the periportal region of the liver (suggested by the other reviewer). We observed that mitochondrial perimeter was reduced in the perivenous region (Figure 1C) but increased mitochondrial perimeter in the periportal (morphologically more round and blotted mitochondria) (Supplementary figure 1B). was observed in the ED-fed mice when compared to CD-fed mice.

No significant changes were observed in the ER perimeter between CD and ED-fed mice in the perivenous region (Figure 1D), meanwhile, the ER perimeter was significantly decreased in the periportal region (Supplementary figure 1C) of the ED-fed mice compared with the CD-fed mice. Interestingly, the percentage of mitochondria in close apposition to the ER was increased in both the perivenous region (Figure 1E) and the periportal region (Supplementary figure 1D). However, a significant increase in the MAM length was observed in the perivenous region (Figure 1G) but not in the periportal region (Supplementary figure 1F), indicating that the reduction of MAM length in the ED-fed periportal region may have resulted due to an increase in the overall mitochondrial perimeter.

Together, these results revealed a variable effect of alcohol on mitochondria and ER ultrastructure in the pericentral and periportal hepatocytes population. Notably, despite having difference in the MAM length, the association between ER and mitochondria was enhanced in both the regions, suggesting alcohol promotes MAM formation but more prominently in the pericentral region.

6- Fig 1C-D-E-G: What does represent one dot in the graph? One cell? In the legends, the authors mentioned only n=3 independent mice. To strengthen this data, is it possible for the authors to precise both number of cells and also mitochondria evaluated approximately in each mice. Furthermore, if they pool the individual data, is it possible to provide a test showing that no differences between mice from the same group?

The no. of points presented in the graph represents the number of electron microscopic fields analyzed. We used number of microscopic fields as “n” number as we couldn’t plot graph using

GraphPad prism software with “n” number more 256 which we have exceeded. To address this concern, we have added the actual number of mitochondria analyzed in the study in the corresponding figure (which is 260 mitochondria for CD and 390 mitochondria for ED in the perivenous region and 160 and 230 mitochondria for CD and ED, respectively, in the periportal region).

7- Supp Table 1 displays an increase of mRNA levels of Hspa9 gene, which encodes for GRP75 (+0.361 log₂FC), but also of Itpr1 gene (+0.573 log₂FC). How the authors can not exclude that EtOH-Mam formation is not the result of this increased expression, independently of GRP75?

- We agree with the reviewer that RNA seq data showed increased mRNA levels of Hspa9 (GRP75) and Itpr1 (IP3R1) but not significant. However, we did not observe any significant difference in total protein expression of these genes as shown in the supplementary figure 6H-J. This indicates that the increase in MCC complex formation by alcohol was not caused by an increase in the expression of IP3R1 or GRP75.

8- Fig 2F. The authors could add a relative quantification of PDK4 protein level as it does not appear to be upregulated according to the blot.

- A relative quantification of the PDK4 protein level of Fig 2F has been added. The quantification of the PDK4 protein levels did not achieve statistical significance (*p value*=0.06) when compared between HS and ALD human samples. However, we observed an increasing trend of PDK4 expression in ALD.

9- Page 3 - Line 22: The authors mention the aberrant increase of MAM and subsequent mitochondrial dysfunction during hepatic insulin resistance. Because of the role of MAM in multiple cellular phenotypes related to hepatic insulin resistance and/or liver functions, autophagy (DOI: 10.3389/fcell.2020.00595), inflammation (DOI: 10.1038/s41419-017-0027-2), apoptosis (DOI: 10.1126/science.1189157) and also cellular senescence (DOI: 10.1038/s41467-021-20993-z). The authors should discuss the consequence of the aberrant induction of MAM on cellular phenotype and how it relates to the liver phenotype they observed.

- As suggested by the reviewer, we have added this information in the revised manuscript (please refer to the 6th paragraph of the discussion section).

10- Page 3 - Line 26.27: The sentence “Whereas, ...high fat diet-induced steatosis” should be corrected.

- This line has been corrected in the revised manuscript.

11- Fig5: The main known function of PDK4 is to phosphorylate and inhibits PDH, reducing mitochondrial Acetyl-CoA generation and subsequent oxidative phosphorylation. Manipulating PDK4 has been shown to modify fatty acid oxidation, but also glycolysis (DOI: 10.1038/s41598-017-09163-8, DOI:10.1016/j.mito.2019.07.009). The authors should discuss that point and tone down the conclusions based on role of PDK4 only in MAM formation.

- We have toned down on our claims and added more relevant information to address the contribution of PDC dependent role of PDK4 (please refer to the last part of the 5th paragraph in the discussion section).

12- The link between the aberrant enhanced MAM/mitochondrial dysfunction and the pathological phenotype of the liver observed in EtOH-fed mice remains unclear, especially related to lipid

accumulation. Can the authors comment on this link between MAM and lipid droplets, related to lipogenesis/lipolysis.

It is a very interesting point raised by the reviewer, the fundamental function of MAM includes phospholipid synthesis besides its critical role in regulating Ca^{2+} transport (Flis VV, Daum G. et al Spring Harb Perspect Biol. 2013). Recently several interesting reports have revealed a completely new angle of lipid droplet formation at MAM. Multiple proteins involved in lipid droplet biosynthesis were found to be localized at MAM such as (mitoguridin2) MIGA2 (Freyre CAC et al Mol cell, 2019; Kim H. et al. Nat Comm, 2022), Seipin (Combot Y et al. Cell rep, 2021), ORP5 and ORP8 (Guyard V et al. J Cell Biol, 2022). Interestingly, ORP5/8 was found to regulate seipin recruitment to the MAM-LD contact site to mediate LD biogenesis. Importantly, the integrity of ER-mitochondria contact sites is crucial for orchestrating LD biogenesis and maturation (Guyard V et al. J Cell Biol, 2022), suggesting that MAM formation may play a role in lipid droplet formation in ALD which requires further investigation.

We added these perspectives in the discussion (please refer to the 5th paragraph in the discussion section).

13- Finally, beyond AST (which is not specific for liver damage, and which is the result also of other tissue damages, such as muscle), ALT and triglycerides levels, the proper liver functions have not been clearly evaluated in ED-fed mice. Do ED-fed mice display hyperglycemia at basal level? Do they present the inability of glucose clearance or some insulin resistance? Could it be possible to evaluate (or at least discuss) the function of these livers? And also reusing it in the some of the different presented conditions?

- Unfortunately, comparing hyperglycemia, glucose clearance or insulin resistance analysis between CD-fed and ED-fed groups was not possible in the current settings as these animals were administered with EtOH or Maltose-Dextrose as isocaloric control on the last day of the experiment for 8 hours before the euthanizing these animals. In addition, for proper study of glucose intolerance, glucose level and insulin sensitivity analysis have to be conducted in a fasted state with insulin injection which was out of the focus of this current study.

REVIEWER COMMENTS

Reviewer #1 (Remarks to the Author):

The results presented in this work provide an important advance in understanding the role of PDK4, GRP75 and MAMs in ALD. The comments made by the reviewers have been thoroughly addressed and the manuscript has been greatly improved.

A minor comment remains. The additional information provided in figure 1 concerning the impact of ethanol on mitochondria perimeter in the perivenous region of the liver suggests network fission. Furthermore, reference 14 used to explain the decrease in mitochondrial respiration under conditions of calcium overload and increased ROS production highlights mitochondrial fission. A discussion of altered mitochondrial dynamics would be of value.

Reviewer #2 (Remarks to the Author):

The rebuttal by the authors raises very serious concerns. Their response indicates they are either very sloppy or disingenuous. Either of which makes the credibility of the manuscript come into question.

1) Response to major point - 1st paragraph (Reviewer 2) - "We were aware of this issue from the beginning of the study, and we have looked at the perivenous region of the liver. Now, we added the description of the region analyzed in the manuscript for the EM figures."

The authors are implying that they originally analyzed perivenous regions of the liver and yet in the original submission they did not state this, but rather stated they analyzed mitochondria in the liver (implying whole liver). They have now simply relabelled Figure 1, as if that is okay. If you analyzed one zone of the liver, you cannot have not mention this in the original manuscript. It is misleading and very wrong. It raises suspicion.

2) Response to major point - 2nd paragraph (Reviewer 2) - "To address this specific concern of the reviewer, we have applied the same parameters (ER perimeter, mitochondria perimeter, % of mitochondria in apposition to ER and MAM length) and analyzed the periportal region."

The authors now claim they have analyzed the periportal region of the liver, yet there is no description of how this was done. One cannot simply look under an electron microscope and determine what type of hepatocyte is being observed. It is very tricky at close magnification. Without any description of methods distinguishing perivenous and periportal hepatocytes by electron microscopy, how can we be certain this was achieved? How can anyone reproduce the work? What about other zones of the liver? Without a detail description of how perivenous and periportal hepatocytes were distinguished by electron microscopy, this data must be questioned.

Reviewer #3 (Remarks to the Author):

vMost of my concerns have been properly addressed by the authors. There is still one minor point that remains for calcium imaging. Beyond that, I found this manuscript now suitable and valuable to be published in Nature Communications.

Figure 3K and Figure 3M. The authors performed the calcium imaging experiments with ratiometric probe and the basal concentration of mitochondria is then accurate. Furthermore, upon ATP stimulation, it would be better to represent the Delta from basal to stimulation peak.

Point-by-point response to the reviewers' comments

REVIEWER COMMENTS

Reviewer #1 (Remarks to the Author):

The results presented in this work provide an important advance in understanding the role of PDK4, GRP75 and MAMs in ALD. The comments made by the reviewers have been thoroughly addressed and the manuscript has been greatly improved.

A minor comment remains. The additional information provided in figure 1 concerning the impact of ethanol on mitochondria perimeter in the perivenous region of the liver suggests network fission. Furthermore, reference 14 used to explain the decrease in mitochondrial respiration under conditions of calcium overload and increased ROS production highlights mitochondrial fission. A discussion of altered mitochondrial dynamics would be of value.

Response:

We have added the following information regarding altered mitochondrial dynamics in the discussion section (3rd paragraph- red text in the revised manuscript).

“MAM plays a significant role in MQC by regulating mitochondrial fission (Friedman et al., 2011). Additionally, alcohol induces mitochondrial fission in hepatocytes (Zhou et al., 2019); in line with our observation, we found that ED reduced mitochondrial perimeter in the hepatocytes of the perivenous region, the primary site affected by alcohol (Guerra et al., 1987). We have recently demonstrated that PDK4 promotes stress-induced mitochondrial fission (Thoudam et al., 2022). Together, these evidences suggest that PDK4 may play a role in mitochondrial fission in coordination with its role in MAM formation in ALD. Of note, in contrast to perivenous hepatocytes, the mitochondrial perimeter was increased in the periportal hepatocytes of ED-fed mice which correlates with an earlier report suggesting that alcohol promotes the formation of megamitochondria (Matsuhashi et al., 1998). These findings suggest that the effect of alcohol on mitochondrial shape and sizes may vary in different zones of the liver but how hepatocytes in different zones react differently in response to alcohol needs further investigation.”

We highly appreciate the reviewer's valuable suggestion during the review process. We believe that addressing all the reviewer's concerns have greatly improved the quality of this manuscript.

Reviewer #2 (Remarks to the Author):

The rebuttal by the authors raises very serious concerns. Their response indicates they are either very sloppy or disingenuous. Either of which makes the credibility of the manuscript come into question.

1) Response to major point - 1st paragraph (Reviewer 2) - "We were aware of this issue from the beginning of the study, and we have looked at the perivenous region of the liver. Now, we added the description of the region analyzed in the manuscript for the EM figures."

The authors are implying that they originally analyzed perivenous regions of the liver and yet in the original submission they did not state this, but rather stated they analyzed mitochondria in the liver (implying whole liver). They have now simply relabelled Figure 1, as if that is okay. If you analyzed one zone of the liver, you cannot have not mention this in the original manuscript. It is misleading and very wrong. It raises suspicion.

2) Response to major point - 2nd paragraph (Reviewer 2) - "To address this specific concern of the reviewer, we have applied the same parameters (ER perimeter, mitochondria perimeter, % of mitochondria in apposition to ER and MAM length) and analyzed the periportal region."

The authors now claim they have analyzed the periportal region of the liver, yet there is no description of how this was done. One cannot simply look under an electron microscope and determine what type of hepatocyte is being observed. It is very tricky at close magnification. Without any description of methods distinguishing perivenous and periportal hepatocytes by electron microscopy, how can we be certain this was achieved? How can anyone reproduce the work? What about other zones of the liver? Without a detail description of how perivenous and periportal hepatocytes were distinguished by electron microscopy, this data must be questioned.

Response:

To study the immediate effect of alcohol on MAM formation, we targeted the perivenous hepatocytes around the central vein (CV) area from the beginning of this study as we (Yang et al. JCI Insight. 2021) and others (Bertola et al. Nat Protocol 2014; Yun et al. FRMB. 2014) have observed high induction of Cyp2E1 (a sensitive marker for alcohol exposure) around this region. Moreover, under the electron microscopy CV was relatively easier to locate. We accept your point that we did not mention this in the original manuscript as we were not trying to compare the effect of alcohol on MAM formation in the different zones of the liver, rather our study, both *in vivo* and *in vitro*, was designed to understand the direct effect of alcohol on MAM dynamics.

We highly appreciate the reviewer's suggestion to compare the zonal difference on MAM dynamics between perivenous and periportal hepatocytes during the first revision. We took this suggestion seriously, as we carefully analyzed the liver sections by EM under low and high magnification to precisely map the CV and periportal vein (PV) regions (as shown in the figures below which was taken during the first revision). We respectfully accept the reviewer's comments that describing the method was necessary to avoid any misinterpretation and we have added this information in this revised version.

The following description has been added in "Transmission electron microscopy (TEM)" part of the materials and methods section:

"To map the periportal and perivenous region in the liver sections, a combination of low- and high-magnification EM imaging ranging from 38X to 5000X was applied to precisely locate the hepatocyte population in the periportal and perivenous region (within 50µm from the CV or PV border) as demonstrated in the supplementary Fig. 1a. The architecture of mitochondria and ER in the EM images were reconstructed graphically using GIMP software (GNU image manipulation program, Gimp 2.10) and analyzed using ImageJ (NIH, Bethesda, MD, USA)."

Mapping of Perivenous and Periportal Region
(performed during the revision)

CV: Central Vein / PV: Portal Vein

Image taken during the revision

Moreover, we also compared the results separately between the data in the original manuscript and data collected during the revision (as shown in the figure below). In both analyses, we found a similar observation (performed in an unbiased manner by our EM experts) that ED-promotes MAM formation in the perivenous hepatocytes. Regarding relabeling of the existing figures, we were assured that the existing images are from the perivenous hepatocytes and as we focused around the CV region while taking these images, some part of the CV is prominently visible in these images as shown in the figures below.

Comparison of old and new data

Figure 1b (Old images)

In this revised manuscript, we have added a detailed description of the methods in the materials and methods section. Additionally, a separate figure, demonstrating how we distinguished and analyzed the perivenous and periportal region in the supplementary Figure 1A.

We again appreciate the reviewer for pointing out this issue. We hope that we have addressed the reviewer's concerns satisfactorily with this additional information. We believe that the overall quality of our manuscript is significantly enhanced, based on these valuable comments.

Reviewer #3 (Remarks to the Author):

Most of my concerns have been properly addressed by the authors. There is still one minor point that remains for calcium imaging. Beyond that, I found this manuscript now suitable and valuable to be published in Nature Communications.

Figure 3K and Figure 3M. The authors performed the calcium imaging experiments with ratiometric probe and the basal concentration of mitochondria is then accurate. Furthermore, upon ATP stimulation, it would be better to represent the Delta from basal to stimulation peak.

Response:

ATP stimulation peak graph has been replaced with the graph of delta from the basal to the stimulation peak. We have added the description about this quantification in its respective figure legends.

We would like thank the reviewer for providing thoughtful critiques on our findings. We believe that addressing all the reviewer's concerns have greatly improved the overall quality of our manuscript.

REVIEWERS' COMMENTS

Reviewer #2 (Remarks to the Author):

The additions to the paper are helpful.

Point-by-point response to the reviewers' comments

REVIEWERS' COMMENTS

Reviewer #2 (Remarks to the Author):

The additions to the paper are helpful.

Response:

We greatly appreciate the reviewer's thoughtful criticism during the review process. We are pleased to have satisfactorily addressed the reviewer's concerns with the additional information.